**Improved methodologies for Earth system modelling of atmospheric soluble iron and observation comparisons using the Mechanism of Intermediate complexity for Modelling Iron (MIMI v.1.0).**

Douglas S. Hamilton[1], Rachel A. Scanza[2], Yan Feng[3], Joe Guinness[4], Jasper F. Kok[5], Longlei Li[1], Xiaohong Liu[6], Sagar D. Rathod[7], Jessica S. Wan[1], Mingxuan Wu[6], and Natalie M. Mahowald[1]

1. Department of Earth and Atmospheric Science, Cornell University, Ithaca, NY, USA
2. Atmospheric Sciences and Global Change Division, Pacific Northwest National Laboratory, Richland, Washington, USA
3. Environmental Science Division, Argonne National Laboratory, Argonne, IL, USA
4. Department of Statistics and Data Science, Cornell University, Ithaca, NY, USA
5. Department of Atmospheric and Oceanic Sciences, University of California, Los Angeles, CA 90095, USA
6. Department of Atmospheric Science, University of Wyoming, Laramie, WY, USA
7. Department of Civil and Environmental Engineering, University of Illinois at Urbana-Champaign, Urbana, IL, USA

## Abstract

Herein, we present the description of the Mechanism of Intermediate complexity for Modelling Iron (MIMI v1.0). This iron processing module was developed for use within Earth system models and has been updated within a modal aerosol framework from the original implementation in a bulk aerosol model. MIMI simulates the emission and atmospheric processing of two main sources of iron in aerosol prior to deposition: mineral dust and combustion processes. Atmospheric dissolution of insoluble to soluble iron is parametrized by an acidic interstitial aerosol reaction and a separate in-cloud aerosol reaction scheme based on observations of enhanced aerosol iron solubility in the presence of oxalate. Updates include a more comprehensive treatment of combustion iron emissions, improvements to the iron dissolution scheme, and an improved physical dust mobilization scheme. An extensive dataset consisting predominantly of cruise-based observations was compiled to compare to the model. The annual mean modelled concentration of surface-level total iron compared well with observations, but less so in the soluble fraction (iron solubility) where observations are much more variable in space and time. Comparing model and observational data is sensitive to the definition of the average and the temporal and spatial range over which it is calculated. Through statistical analysis and examples, we show that a median or log-normal distribution is preferred when comparing with soluble iron observations. Comparison of iron solubility calculated at each model time step versus that calculated based on a ratio of the monthly mean values, which is routinely presented in aerosol studies and used in ocean biogeochemistry models, are on average globally one-third (34%) higher. We redefined ocean deposition regions based on dominant iron emission sources and found that the daily variability in soluble iron simulated by MIMI was larger than that of previous model simulations.

MIMI simulated a general increase in soluble iron deposition to Southern Hemisphere oceans by a factor of two to four compared with the previous version, which has implications for our understanding of the ocean biogeochemistry of these predominantly iron limited ocean regions.

## 1 Introduction

Iron is an essential micronutrient for ocean primary productivity (Martin et al., 1991; Martin, 1990). Iron deficiency in oceans leads to high-nutrient low-chlorophyll (HNLC) conditions under which the photosynthetic productivity of phytoplankton is iron limited (Boyd et al., 2007; Jickells et al., 2005), and in other regions iron may be an important nutrient for nitrogen fixation by diazotrophs (Capone et al., 1997; Moore et al., 2013, 2006). Atmospheric deposition of bioavailable iron (i.e., the fraction of the total iron deposited that is readily available for ocean biota uptake) contained in aerosol is an important source of new iron for the remote open ocean (Duce and Tindale, 1991; Fung et al., 2000); therefore, iron impacts the ability of oceans to act as a sink of atmospheric carbon dioxide (Jickells et al., 2014; Moore et al., 2013).

Several definitions for bioavailable iron have been proposed. The solubility of iron is considered to be a key factor modulating its bioavailability (Baker et al., 2006a, 2006b); therefore, we consider bioavailable iron to be the dissolved (labile) iron in either a (II) or (III) oxidation state, and we define this as the soluble iron concentration throughout the manuscript. However, since most aerosol iron is insoluble at emission the processing of insoluble iron to a soluble form must occur during atmospheric transport. The acidic processing of iron contained in aerosol is one pathway under which soluble iron can be liberated from an insoluble form with decreasing pH (Duce and Tindale, 1991; Solmon et al., 2009; Zhu et al., 1997). Organic ligands, in particular oxalate, also increase iron solubility by weakening or cleaving the Fe–O bonds found in iron oxide minerals via complexation (Li et al., 2018; Panias et al., 1996), and in nature this reaction proceeds most rapidly in a slightly acidic aqueous medium, such as cloud droplets (Cornell and Schindler, 1987; Paris et al., 2011; Xu and Gao, 2008). Organic ligand processing has been estimated to increase soluble iron concentrations by up to 75% more than is achievable with acid processing alone (Ito, 2015; Johnson and Meskhidze, 2013; Myriokefalitakis et al., 2015; Scanza et al., 2018). However, there is no single mechanism that describes the observed inverse relationship of higher iron solubilities with decreasing iron concentrations (Sholkovitz et al., 2012). Rather, Mahowald et al. (2018) used a 1–D plume model to demonstrate that the observed trend can be explained by either the differences in iron solubility at emission or the atmospheric dissolution of insoluble iron.

Thus, there is no observational constraint to indicate which is more likely, unless spatial
distribution is also considered.
The recent increase in efforts to model iron solubility (Ito, 2015; Ito and Xu, 2014; Johnson and
Meskhidze, 2013; Luo et al., 2008; Meskhidze et al., 2005; Myriokefalitakis et al., 2015; Scanza
et al., 2018) reflects its importance for understanding biogeochemical cycles (Andreae and
Crutzen, 1997; Arimoto, 2001; Jickells et al., 2005; Mahowald, 2011) and how human activity may
be perturbing them (Mahowald et al., 2009, 2017). However, the multi-faceted nature of how iron
interacts within the Earth system results in many uncertainties regarding how best to represent
the atmospheric iron cycle within models, which are themselves of varying complexity
(Myriokefalitakis et al., 2018). To incorporate the processes currently thought to be the most
significant (Journet et al., 2008; Meskhidze et al., 2005; Paris et al., 2011; Shi et al., 2012) and
improve model-to-observation comparisons of the soluble iron fraction, particularly in remote
ocean regions (Baker et al., 2006b; Ito, 2015; Mahowald et al., 2018; Matsui et al., 2018;
Sholkovitz et al., 2012), model development has been focused on refining the atmospheric iron
emission sources and subsequent atmospheric processing (Ito, 2015; Ito and Xu, 2014; Johnson
and Meskhidze, 2013; Luo et al., 2008; Meskhidze et al., 2005; Myriokefalitakis et al., 2015;
Scanza et al., 2018).
A recent multi-model evaluation of four global atmospheric iron cycle models (Myriokefalitakis et
al., 2018) showed that total iron deposition is over-represented close to major dust source regions
and under-represented in remote regions compared with observations from all four models. This
is consistent with previous model inter-comparison studies that demonstrated the difficulty in
simultaneously simulating both atmospheric concentrations and deposition fluxes of desert dust
(Huneeus et al., 2011). Importantly, none of the atmospheric iron processing models can capture
the high (>10%) solubilities measured over the Southern Ocean; this is potentially owing to the
model processes associated with transport and aging of aerosol iron requiring further
development (Ito et al., 2019). Conclusions from Myriokefalitakis et al. suggest that future model
improvements should focus on a more realistic aerosol size distribution and the representation of
mineral-to-combustion sources of iron. Most of the development of the Mechanism of Intermediate
Complexity for Modelling iron (MIMI), as described herein, focused on these points. First, we
transitioned from a bulk aerosol scheme to a two-moment modal aerosol scheme (Liu et al., 2012),
and second, we re-evaluated pyrogenic iron emissions from anthropogenic combustion and fires.
The modal aerosol scheme was used to calculate both aerosol mass and number at each time
step within an updated global aerosol microphysics model, and both the fire and anthropogenic
combustion emissions from Luo et al. (2008), which are likely to be underestimated (Conway et
al., 2019; Ito et al., 2019; Matsui et al., 2018), were improved upon.

Ocean observations of iron, and its soluble fraction, are limited both spatially and temporally owing
to the significant costs and logistical constraints associated with accumulating data from scientific
cruises. Thus, there is an inherent disparity in attempting to compare climatological means
calculated from temporally-chronological model results with observational means calculated from
temporally-limited and sporadic observations (e.g., Mahowald et al., 2008, 2009). This is
important because natural aerosol emissions are variable on seasonal, annual, and decadal
timescales, both in terms of primary natural iron emission sources (mineral dust and wildfires)
and the source of aerosol acidity. For example, sulphuric acid from the oxidation of dimethyl
sulphide and fire $SO_2$ (Bates et al., 1992; Chin and Jacob, 1996) have been observed to aid iron
dissolution when far from anthropogenic acid sources (Zhuang et al., 1992). Limitations
associated with the collection of continuous annual or inter-annual ship-based data across
multiple remote ocean regions are immutable at present, which hinders the required derivation of
basic statistical properties of such highly-variable data (Smith et al., 2017). Attention could
therefore be given to the methodologies under which such model-observation comparisons are
undertaken instead.
The present manuscript is presented in four parts. The first part (section 2) introduces updates
made to the Bulk Aerosol Module (BAM) iron scheme of Scanza et al. (2018) and its
implementation within the Modal Aerosol Module (MAM), with four modes (MAM4), within the
Community Earth System Model (CESM). In the second part (section 3), we compare iron
concentrations and the fractional solubility of iron with the observational data. Then the third part
(section 4) compares our updated version of the model with its predecessor. Finally, we suggest
further developments for atmospheric iron modelling and for comparing model results with
sporadic observations (section 5).


**2 Aerosol model**
The present study improves upon the previous atmospheric iron cycle module developed for the
Community Atmosphere Model (CAM) version 4 (CAM4) embedded in the CESM; we will refer to
this version as BAM-Fe (Scanza et al., 2015, 2018) therein. We incorporated the iron module
within the MAM framework (Liu et al., 2012, 2016) currently in the Department of Energy's Energy
Exascale Earth System Model (E3SM; Golaz et al., 2019) and the CAM versions 5 and 6 (CESM-
CAM5/6; (Neale et al., 2010)); we refer to this new version of the iron model by its name (MIMI)
therein . Table 1 serves as a refence and summarizes the modifications made for MIMI, which
are discussed throughout the manuscript.
We use MAM4 with four simulated log-normal aerosol size modes: three modes (Aitken,
accumulation, and coarse) containing iron and a fourth primary carbonaceous mode. Table 2
details the new pyrogenic iron (i.e., from fires and anthropogenic combustion) modal aerosol
properties, while those of mineral dust iron follow existing dust aerosol properties (Liu et al., 2012).
Generally, the modelled density of iron is similar to size-resolved ambient aerosol densities
measured in Eastern China (Hu et al., 2012), which has significant dust and combustion aerosol
sources. MIMI was initially implemented and tested within a development branch of CAM 5.3, as
per Wu et al. (2017) and Wu et al. (2018), using Cheyenne (Computational and Information
Systems Laboratory, 2017) and closely resembles CESM version 1.2.2. We used a 2.5° x 1.9°
horizonal (longitude by latitude) resolution and 56 vertical layers up to 2 hPa. Stratiform
microphysics followed a two-moment cloud microphysics scheme (Gettelman et al., 2010;
Morrison and Gettelman, 2008). The other major aerosol species black carbon (BC), organic
carbon, sea salt and sulphate ($SO_4$) were also simulated but are not explicitly examined here
because we are focused on iron aerosol modelling. However, atmospheric iron processing in MIMI
requires both sulphate and (secondary) organic aerosols to be simulated as they act as proxies
for the reactant species of [H+] and oxalate, respectively. In CAM5 sulphate aerosol is present in
all three aerosol modes while secondary organic aerosol is only present in the fine Aitken and
accumulation modes (Liu et al., 2012, 2016). Aerosol microphysics was applied in the same way
to the new iron aerosol tracers as the base aerosol species (Liu et al., 2012, 2016). Fire emissions
were vertically distributed between six injection height ranges: 0–0.1, 0.1–0.5, 0.5–1.0, 1.0–2.0,
2.0–3.0, and 3.0–6.0 km, as per AeroCom recommendations (Dentener et al., 2006). Fire
emissions were uniformly distributed in model levels between height limits. Unless otherwise
stated, aerosol and precursor gas mass emissions were from the Climate Model Intercomparison
Program (CMIP5) inventory (Lamarque et al., 2010). Major gas-phase oxidants ($O_3$,OH, $NO_3$ and
$HO_2$) were supplied offline and were also from Lamarque et al. (2010). Meteorology (*U*, *V*, and *T*)
was nudged to Modern-Era Retrospective analysis for Research and Applications (MERRA) data
for 2006-2011. Unless otherwise stated, the last five years were used for analysis.


**Table 1**. Short summary of major differences between BAM-Fe and MIMI.

| BAM-Fe (CAM4)<br>Externally mixed bulk aerosol tracers<br>with 4 size bins<br>(0.1-1.0, 1.0-2.5, 2.5-5.0, 5.0-10.0 µm) | MIMI (CAM5)<br>Internally mixed 2-moment aerosol tracers<br>with 3 aerosol iron size modes<br>(Aitken, accumulation, coarse) |
|---|---|
| Static soil erodibility from offline maps:<br>DEAD (Zender et al., 2003) scheme | Time-varying soil erodibility calculated online:<br>Kok et al. (2014a) scheme |
| 8 dust minerals, 5 of which are iron bearing | No change |
| Static Luo et al. combustion iron emissions | Static Luo et al. combustion iron emissions x5 |
| Static Luo et al. fire iron emissions | Time-varying Fe:BC fire iron emission ratio |
| Surface fire iron emissions | Vertically distributed fire iron emissions |
| Static aerosol pH across aerosol size bins | Aerosol pH size dependent |
| Assumed oxalate concentration based on<br>primary organic carbon | Assumed oxalate concentration based on<br>secondary organic carbon |
| In-cloud aerosol concentrations based on<br>simulated cloud fraction | Separate in-cloud and interstitial aerosol<br>tracers |



The model used in this study performed well when compared to observations from a variety of
different environments, and produced aerosol concentrations that were close to those of the multi-
model mean of similarly complex aerosol models (Fanourgakis et al., 2019).

**Table 2.** Combustion iron aerosol size and number properties.

| Mode | Number mode diameter, $D_{gn}$ (µm) | Geometric standard deviation ($\sigma$) | Volume mean particle diameter, $D_{emit}$ (µm)[1] | Density, $\rho$ (kg/m$^3$) |
|---|---|---|---|---|
| Aitken | 0.03[a] | 1.8[a] | 0.0504 | 1500[c] |
| Accumulation | 0.08[a] | 1.8[a] | 0.134 | 1500[c] |
| Coarse | 1.00[b] | 2.0[b] | 2.06 | 2600[c] |

1. $D_{emit} = D_{gn} \times \exp(1.5 \times (\ln(\sigma))^2)$
a. Liu et al. (2012)
b. Dentener et al. (2006) and Liu et al. (2012)
c. Wang et al. (2015)

## 2.1 Dust aerosol modelling

Mineral dust aerosol was modelled via the Dust Entrainment And Deposition model (DEAD; Zender et al., 2003), which was previously updated to include the brittle fragmentation theory of vertical dust flux (Kok, 2011) on mineral size fractions (Albani et al., 2014; Scanza et al., 2015). We further improved the emissions of dust in MAM to follow a physically-based vertical flux theory (Kok et al., 2014a), which has been shown to significantly improve dust emissions (Kok et al., 2014b). Notice that this method allowed for the removal of the soil erodibility map approach previously employed by the DEAD scheme (Table 1), and still provided more accurate simulations of regional dust emissions and concentration (Kok et al., 2014b). Dust aerosol optical depth (AOD) was calculated using mineralogy-based radiation interactions as described by Scanza et al. (2015). Dust emissions were tuned such that a global annual mean dust AOD of ~0.03 was attained, as recommended by Ridley et al. (2016) and matching values in Scanza et al. (2015) for a similar model configuration.

Dust minerology in MIMI is designed to be comprised of eight separate transported tracers: illite, kaolinite, montmorillonite, hematite, quartz, calcite, feldspar and gypsum (Scanza et al., 2015). Mineral soil distributions were supplied offline (Claquin et al., 1999) with the emission of each dust mineral species further refined following the brittle fragmentation theory (Scanza et al., 2015).

## 2.2 Iron aerosol modelling

The simulated lifecycle of iron can be grouped into three main stages: (1) iron emission to atmosphere, (2) physical-chemical iron processing during transport and (3) final iron deposition and, thus, loss from the atmosphere. In the following sections, we describe the emissions and subsequent atmospheric dissolution of iron (stages 1 and 2), while the effects of this on the magnitude of oceanic soluble iron deposition (stage 3) in MIMI are examined and compared to BAM-Fe in section 4.

Iron optical properties are currently considered to reflect those of hematite because this mineral contains 97% of the iron aerosol mass fraction (see section 2.3.1).

**2.3 Iron aerosol emissions**

MIMI contains three major iron emission sources: mineral dust, fires (defined here as the sum of wildfires and human-mediated biomass burning) and anthropogenic combustion (defined here as the sum of industrial and domestic biofuel burning). In the BAM-Fe version of the model, fire and anthropogenic combustion emissions were combined into a single static monthly mean value. In MIMI, fire emissions of iron were updated to be distinct from other pyrogenic iron sources and were parametrized to track the BC emissions from fires using an Fe:BC ratio. Fire BC emissions were simulated to be time varying on a monthly scale, resulting in a much more pronounced seasonality to fire emissions (e.g., Giglio et al., 2013) compared to BAM-Fe where seasonality was not imposed.

For all iron species in each mode, the aerosol number emissions ($Fe_{emit,num}$) were calculated from the mass emissions within the same mode ($Fe_{emit,mass}$) using the properties in Table 2 and following Liu et al. (2012),

$$Fe_{emit,num} = \frac{Fe_{emit,mass}}{\left(\frac{\pi}{6}\right) \times \rho \times D_{emit}^3}$$
Equation 1

**2.3.1 Iron emissions within mineral dust aerosol**

Based on previous research by Journet et al. (2008) and Ito and Xu (2014), the iron fraction in each mineral species was prescribed at emission as follows: 57.5% in hematite, 11% in smectite, 4% in illite, 0.24% in kaolinite, 0.34% in feldspar, and 0% in the remaining three mineral species (Table 3); which has been shown to improve the accuracy of the modelled total iron fraction estimated from mineral dust (Scanza et al., 2018; Zhang et al., 2015). The mass of each of the eight mineral dust species advected at each model time step was the residual mineral mass (i.e., after the removal of the iron mass), such that the sum of all eight minerals and the total iron from mineral dust equalled unity, and hence, the original total singular dust mass emitted from the land surface.

Iron emissions from the five iron-bearing mineral dust species (three dust minerals contain no iron) were then partitioned into the four advected mineral-dust-bearing iron aerosol tracers (Table 3); iron tracers were defined as being (in)soluble and by the speed of the atmospheric reaction rate acting on them: slow or medium (Scanza et al., 2018). Note that, slow- and med-soluble iron are only produced by non-reversable atmospheric processing within the model; therefore,

computational costs can be reduced by not creating a separate iron tracer representing the
fraction which is already soluble at emission (i.e., 'fast' reacting), but instead add an initial med-
soluble iron processed emission burden which is equivalent to the assumed fast reacting iron
fraction.

**Table 3.** Mass fraction of iron in each simulated iron bearing dust mineral species and allocation
to each mineral iron tracer at emission. At emission med-soluble iron is equivalent to the fast-
soluble iron fraction (i.e., the fraction which is already assumed to be soluble at emission).
Residual mineral dust mass is then advected as its respective tracer.

| | Mineral dust mass percent allocated to each dust iron tracer at emission | | | | |
|---|---|---|---|---|---|
| Mineral | Med-soluble | Med-insoluble | Slow-soluble | Slow-insoluble | Total |
| Hematite | 0.0% | 0.0% | 0.0% | 57.5% | 57.5% |
| Smectite | 0.55% | 10.45% | 0.0% | 0.0% | 11.0% |
| Illite | 0.11% | 3.89% | 0.0% | 0.0% | 4.0% |
| Kaolinite | 0.01% | 0.0% | 0.0% | 0.23% | 0.24% |
| Feldspar | 0.01% | 0.0% | 0.0% | 0.33% | 0.34% |



### 2.3.2 Iron aerosol emissions from fires

Following Luo et al. (2008), we used observed Fe:BC mass ratios to estimate fine and coarse
mode iron emissions from fires. An additional difference between BAM (CAM4) and MAM (CAM5)
models is the emission dataset used to estimate global fire emissions of aerosol and trace gases.
The BAM model uses adjusted AeroCom fire emissions (Dentener et al., 2006; Scanza et al.,
2018), while MAM uses CMIP5 fire emissions (Lamarque et al., 2010). Base fire BC emissions
within the CMIP5 database are 2.55 Tg a$^{-1}$ BC; however, the scaling of emissions from fires has
been shown to be necessary to improve model to observed (aerosol optical depth and particulate
matter) BC ratios (Reddington et al., 2016; Ward et al., 2012). Therefore, we globally scaled the
fire iron emissions by a uniform factor of two, which is comparable with the overall lower scaling
factor from a review of the literature by Reddington et al. (2016: Table 2). Fine mode iron
emissions from fires were then segregated to assign 10% of the fine sized mass to the Aitken
mode, with the remaining 90% assigned to the accumulation mode.

**Table 4.** Measured iron (Fe) and black carbon (BC) values (various units; as only the Fe:BC ratio is required they are not included) and the Fe/BC ratio. Calculated with three decimal places, ratio reported to one significant figure to reflect high uncertainty. Modelled fire emission ratio for Fe:BC then calculated from observed ratios.

272

| Biome | Reference | Fe | BC | Fe/BC |
|---|---|---|---|---|
| Cerrado | Yamasoe et al. (2000) | 0.08 | 12.6 | 0.006 |
| | Yamasoe et al. (2000) | 0.05 | 6.5 | 0.008 |
| | Ward et al. (1991) | 0.9 | 3.3 | 0.273 |
| | Mean Fe:BC ratio = 0.1 | | | |
| Temperate | Ward et al. (1991) | 0.1 | 5.0 | 0.020 |
| | Mean Fe:BC ratio = 0.02 | | | |
| Tropical | Luo et al. (2008) | - | - | 0.020 |
| | Artaxo et al. (2013) | 179 | 2801 | 0.639 |
| | Artaxo et al. (2013) | 27 | 405 | 0.067 |
| | Artaxo et al. (2013) | 20 | 98 | 0.204 |
| | Artaxo et al. (2013) | 12 | 235 | 0.051 |
| | Ward et al. (1991) | 0.9 | 10 | 0.090 |
| | Yamasoe et al. (2000) | 0.03 | 7.3 | 0.004 |
| | Yamasoe et al. (2000) | 0.05 | 3.9 | 0.013 |
| | Mean Fe:BC ratio = 0.06 | | | |
| Global | Mean Fe:BC ratio = 0.06 | | | |

273

274

Luo et al. (2008) used a single Amazonian observational dataset in their study to determine the flux of iron aerosol from fires (Fe:BC). We extended this to incorporate other Amazonian fire (Fe:BC) data and, importantly, non-Amazonian biome fire (Fe:BC) data, which are likely to have different combustion properties, and hence iron emissions (e.g., Akagi et al., 2011). From Table 4, we suggest that after adding 11 more data inventory values, Luo et al. likely under-represented the global fine mode Fe:BC ratio at 0.02. We instead used the global mean Fe:BC ratio from the additional data of 0.06. Conversely, Luo et al. likely over-represented the coarse mode Fe/BC ratio at 1.4. By including additional observational information from Artaxo et al. (2013) we reduced

this to 1.0. Using size-segregated wet season (i.e., representing a locally-transported emission
source) observation data from Artaxo et al. (2013), we estimated that the amount of BC mass in
the coarse mode was 37% of fine mode mass. Overall this doubles the fractional contribution of
fine mode (BAM: 0.1–1µm size bin, MAM: sum of Aitken and accumulation modes) iron emissions
from fires (BAM-Fe: fine = 7% of total mass, MIMI: fine = 14% of total mass).
Using the soluble Fe:BC ratio of 0.02 reported in Luo et al. (2008) resulted in 33% solubility of
fine mode iron from fires at emission, which is lower than the 46% reported in Oakes et al. (2012)
and higher than the 12% reported in Ito (2013). As few data exist in the literature pertaining to
coarse mode BC, or more importantly its ratio to iron, we retained the 4% solubility of iron in the
coarse mode at emission, as suggested by Luo et al.
Total iron emissions from fires in MIMI were 2.2 Tg Fe a$^{-1}$ (Aitken: 0.02 Tg a$^{-1}$, accumulation: 0.28
Tg a$^{-1}$, coarse: 1.9 Tg a$^{-1}$), representing an approximate increase in iron emissions from fires of
around 25% compared with those from BAM-Fe, with most of the mass (86%) still in the coarse
mode. The lower 25% increase between BAM-Fe and MIMI iron emissions, as compared to the
doubling of the fire iron emissions themselves within MIMI, is due to different underlying fire
emission inventories used in each model. Aerosol number concentrations were then calculated
using Equation 1 and the physical properties listed in Table 2. We adopted the methodology of
Wang et al. (2015) by assuming that the density of iron aerosol from fires (and anthropogenic
combustion) in the Aitken and accumulation modes matches that of BC, while in the coarse mode
matches that of mineral dust. The vertical distribution of iron emissions from fires were also
updated in MIMI (BAM-Fe emitted all iron from fires at the surface) to account for pyro-convection,
which lofts aerosol to higher altitudes at the point of emission within the model (Rémy et al., 2017;
Sofiev et al., 2012; Wagner et al., 2018).

### 2.3.3 Iron emissions from anthropogenic combustion sources


Separate lines of evidence (Conway et al., 2019; Ito et al., 2019; Matsui et al., 2018) have shown
that anthropogenic industrial iron emissions are highly likely to be larger than previously estimated
(e.g., Ito, 2015; Luo et al., 2008; Myriokefalitakis et al., 2018a). Therefore, anthropogenic
combustion emissions of iron in MIMI were the same as those in BAM-Fe, as first reported by Luo
et al. (2008), uniformly multiplied by a factor of five to bring into closer agreement with
observations of industrial magnetite emissions in line with Matsui et al. (2018). Resulting fine
mode anthropogenic combustion emissions were 0.50 Tg Fe a$^{-1}$ and coarse mode emissions were
2.8 Tg Fe a$^{-1}$. Similar to fire emissions, 10% of fine size emissions were partitioned into the Aiken
mode at emission, the remainder 90% of fine size emissions were emitted into the accumulation
mode, and 100% of coarse size emissions were emitted to the coarse mode. We retain the Luo
et al. (2008) estimate of 4% combustion iron solubility at emission (Chuang et al., 2005).
Calculations of aerosol number concentrations of combustion iron followed the same procedure
as described for fire emissions in the previous Section 2.3.2.

**2.4 Atmospheric iron aerosol processing**
**2.4.1 Acid and organic ligand processing**
Once airborne, iron undergoes a series of physical and chemical processing steps within the
atmosphere, each working to alter the soluble iron fraction (i.e., its solubility). The MIMI
atmospheric iron dissolution scheme is presented in Table 5, with a full description reported
previously by Scanza et al. (2018). Within each of the three iron-bearing aerosol size modes, six
tracers of iron were advected within the model: medium-insoluble and medium-soluble mineral
dust iron (containing both readily-released and medium-reactive mineral dust iron (Scanza et al.,
2018)), slow-insoluble and slow-soluble mineral dust iron, and insoluble and soluble pyrogenic
(sum of fires and anthropogenic combustion) iron which was assumed to be medium-reactive
(Scanza et al., 2018). Both proton and organic ligand promoted iron dissolution mechanisms were
modelled. The proton promoted dissolution scheme was dependent upon an estimated [H$^+$],
calculated from the ratio of sulphate to calcite, and the simulated temperature. Organic ligand
dissolution was dependent upon the simulated organic carbon concentration as oxalate (the main
reactant) itself was not modelled. Both the sulphate and secondary organic carbon aerosol (Fig.
S1), upon which the iron processing requires, are fundamental components of aerosol models
(e.g., Kanakidou et al., 2005; Mann et al., 2014). In CAM sulphate is mainly formed via oxidation
of SO$_{2(aq)}$ with a smaller contribution from H$_2$SO$_4$ condensation on aerosol while secondary organic
aerosol is formed via the partitioning of semi-volatile organic gases (Liu et al., 2012). Neither gas-
to-particle production processes are structurally modified from the description of CAM5 by Lui et
al. (2012, 2016) by the incorporation of MIMI. A structural model improvement was that MAM
(CAM5) advected separate tracers for the interstitial and cloud-borne aerosol phases, and so the
proton and organic ligand promoted dissolution reactions were applied to each aerosol phase,
respectively.
Dust aerosol moving through areas containing acidic gases, with a pH 1–2, increases the solubility
of the iron contained within it (Ingall et al., 2018; Longo et al., 2016; Meskhidze et al., 2003;
Solmon et al., 2009); with minerology being a key factor determining the rate of dissolution at a
given pH (Journet et al., 2008; Scanza et al., 2018). Modelled aerosol pH in MIMI was
parametrised to depend only on the ratio of the calcium to sulphate aerosol concentration (Scanza
et al., 2018). At each time step, if $[SO_4] > [Calcite]$, then the aerosol was assumed to be acidic
with a low pH, while if $[SO_4] < [Calcite]$, then aerosol was assumed to be well buffered (Böke et
al., 1999) and the pH = 7.5. In MIMI, we updated the pH calculation from BAM-Fe two-fold: (1) In
BAM-Fe, pH was calculated as the mean across all four size bins (0.1–10 µm), while in MIMI, pH
was calculated separately for each interstitial aerosol size mode. (2) Aerosol measurements of
pH have shown that interstitial aerosol is likely to be more acidic than was assumed in BAM-Fe
(Longo et al., 2016; Weber et al., 2016), even when taking into account declining sulphate levels
(Weber et al., 2016); therefore, we have lowered the aerosol pH to 1 (from 2) in both the Aitken
and accumulation modes where sulphate aerosol dominates. However, in the coarse mode,
where dust dominates, we retained the lower pH boundary of 2. Furthermore, MAM aerosol was
simulated as an internally mixed aerosol; therefore, the $SO_4$:Ca ratio included the mixing of these
aerosol components within each mode. See Section 4.2 for comparison of acid processing in
MIMI with literature and previous model (BAM-Fe).
All aerosol species in the host CAM5 framework are carried in either an interstitial (i.e., not
associated with water) or cloud-borne (i.e., associated with water) phase. The organic-ligand
reaction only proceeds within MIMI if the condition that cloud is present in the grid-cell is first met.
If cloud is present then only the iron aerosol which is associated with water undergoes organic
ligand processing (i.e., the interstitial aerosol component remains unchanged). Any future
development of MIMI within an aerosol model which does not advect a separate tracer for the
cloud-borne phase of aerosol would therefore need to adjust the reaction to take account of this.
An assumed oxalate concentration in MIMI was estimated based on the modelled organic carbon
concentration and could not exceed a maximum concentration threshold of 15 µmol/L (Scanza et
al., 2018). In BAM-Fe, oxalate was derived from the sum of both the primary and secondary
organic carbon aerosol concentrations, while in MIMI this was updated to be dependent only upon
the secondary organic carbon source because oxalate is itself a product of the oxidation of volatile
organic carbon gases (Myriokefalitakis et al., 2011). An additional term was added to the reaction
mechanism to account for the small amount of organic ligand processing proceeding by species
other than oxalate (Scanza et al., 2018). See Section 4.2 for comparison of in-cloud organic
dissolution in MIMI with literature and previous model (BAM-Fe).
**Table 5.** Summary of atmospheric processing reaction equations from Scanza et al. (2018). Here
*l* represents either medium or slow reacting iron aerosol (combustion iron is modelled as medium).
The pH calculation is updated to be calculated within each mode and oxalate ($C_2O_4^{2-}$)
concentrations are calculated based only on the secondary organic aerosol (SOA) concentrations.

| Reaction equation | Reaction rate constituents |
|---|---|

**Acid processing of aerosol**

$$RFe_{l,acid} = K_l(T) \times a(H^+)^{m_l} \times f(\nabla G_r) \times A_l \times MW_l$$

Equation 2:

$$\frac{d}{dt}[Fe_{soluble}] = RFe_{i,acid} \times [Fe_{insoluble}]$$

Equation 3:

$$\frac{d}{dt}[Fe_{insoluble}] = -\left(\frac{d}{dt}[Fe_{soluble}]\right)$$

$K_l(T)$ *is the temperature dependent rate coefficient (moles m$^{-2}$ s$^{-1}$)*

$K_{med}(T) = 1.3x10^{-11} \times e^{6.7x10^3 \times (\frac{1.0}{298.0} - \frac{1.0}{temp(K)})}$

$K_{slow}(T) = 1.8x10^{-11} \times e^{9.2x10^3 \times (\frac{1.0}{298.0} - \frac{1.0}{temp(K)})}$

$a(H^+)$ *is the proton concentration, with an empirical reaction order $m_l$*
*$m_{med}$ = 0.39; $m_{slow}$ = 0.50*

*If [SO$_4$] > [Calcite] then pH = 1 in Aitken and accumulation modes or 2 in coarse. Else pH = 7.5.*

$f(\nabla G_r)$ *accounts for dissolution rate change with variation from equilibrium (equals 1 for simplicity (Luo et al., 2008))*

$A_l$ *is the specific surface area (m$^2$ g$^{-1}$)*
$MW_l$ *is the molecular weight (g mol$^{-1}$)*
*$A_{med}$ = 90.0 m$^2$ g$^{-1}$; $A_{slow}$ = 100.0 m$^2$ g$^{-1}$*

**Organic ligand processing**

Equation 4:

$$\frac{d}{dt}[Fe_{soluble}] = RFe_{i,oxal} \times [Fe_{insoluble}]$$

Equation 5:

$$\frac{d}{dt}[Fe_{insoluble}] = -\left(\frac{d}{dt}[Fe_{soluble}]\right)$$

$$RFe_{l,oxal} = a_l \times [C_2O_4^{2-}] + b_l$$

*If l = medium (or combustion) iron:*
*a = 2.3x10$^{-7}$ $\mu$M$^{-1}$ s$^{-1}$; b = 4.8x10$^{-7}$ s$^{-1}$*

*If l = slow iron:*
*a = 9.5x10$^{-9}$ $\mu$M$^{-1}$ s$^{-1}$; b = 3.0x10$^{-8}$ s$^{-1}$*

*For longitude(i), latitude(j) and level(k):*

$$[C_2O_4^{2-}]_{i,j,k} = 150 \times \frac{[SOA_{i,j,k}]}{max[SOA]}$$

## 2.4.2 Computational costs

Earth System models are generally characterized by having a heavy computational burden in simulating atmospheric processes. The inclusion of MIMI requires eight dust mineral tracers (a net addition of seven) and six iron tracers. The total addition of aerosol tracers new is 39 (13 in each of the three aerosol modes) if dust minerology is not already present, or 18 new aerosol tracers if it is (e.g., NASA GISS model (Perlwitz et al., 2015a, 2015b)). The additional computational cost of MIMI within CESM-CAM5 is approximately a doubling of the required core-hours; around half of that is associated with dust minerology speciation and the other half with iron speciation and processing (Table 6). Note that additional computational tuning, or changes in configuration, could modify these computational change estimates. For example, with dust minerology (MAM4DU8) there is an approximate 3-fold increase in required core-hours due to model structural differences when transitioning from CAM5 to CAM6.

Table 6: Simulation time (in seconds per simulated year) for the CESM-MAM4 model. The CAM5 base model, with the addition of dust minerology, and with the addition of dust minerology and iron processing (i.e., MIMI v1.0) shown in black text. Cost of running the new higher resolution CAM6 model with dust minerology also shown for comparison in blue text. All CAM5 simulations executed on 10 nodes, with 36 cores per node, for two years (2006-2007) with consistent output fields.

| | CAM5 | | | CAM6 |
|---|---|---|---|---|
| | MAM4 (Base model) | MAM4DU8 (dust minerology) | MAM4DU8FE6 (MIMIv1.0) | MAM4DU8 (dust minerology) |
| Number advected aerosol species | 24 | 45 | 63 | 46 |
| Gridcell resolution (#lon x #lat) | 144x96 | 144x96 | 144x96 | 288x192 |
| Wall clock s a$^{-1}$(simulation) | 3954 | 5856 | 7836 | 20167 |
| Core-hours | 396 | 586 | 784 | 2017 |

**2.5 Observation and model iron calculations**

**2.5.1 Spatially aggregating limited observations**

The observations of total iron concentrations and the fractional solubility of iron used in this study are the joint totals (1524 records) of those reported in Mahowald et al. (2009) and Myriokefalitakis et al. (2018). However, many of these observations represent averages of only one or a few days of iron and soluble iron measurements, and thus can be difficult to compare against annual, or longer, mean time periods calculated within the model. Furthermore, building empirical distributions of iron properties from observations requires a larger sample size than currently available in many regions. We therefore tested how aggregating the observations spatially, sometimes termed 'super-obbing', altered our model evaluation. Our objective was to capture the small regional scale properties of iron, and not those at a point source; therefore, we assume that the benefits gained by aggregating in this way, to help produce a statistically useful amount of observations, outweighs any potential biases.

**2.5.2 Variations in model temporal averaging**

The model was run at a 30-minute time resolution. At each 30-minute time step, soluble iron, total iron, and the ratio of soluble to total iron (iron solubility) were computed. The model output was $S_i$, (daily mean soluble iron concentration on day $i$), $T_i$ (daily mean total iron concentration on day $i$), and $R_i$ (daily mean iron solubility on day $i$). Note that $R_i$ is the daily mean of the calculated 30-minute solubilities and hence is not equal to $S_i / T_i$. We define online solubility as the average-of-ratios and was calculated as follows:

$$(\sum_{i=1}^{n} R_i)/n \qquad\qquad \text{Equation 6}$$

where $n$ represents the total number of records over which the average was calculated. Online solubility is reported throughout this study. In Section 3.4, we then compare the average-of-ratios to the ratio-of-averages (defined as offline solubility), calculated as follows:

$$\frac{(\sum_{i=1}^{n} S_i)/n}{(\sum_{i=1}^{n} T_i)/n} = \frac{\bar{S}}{\bar{T}} \qquad\qquad \text{Equation 7}$$

where $\bar{S}$ and $\bar{T}$ are the grid cell averages of soluble and total iron concentrations, respectively,
over the total time period considered in this study (2007 to 2011). While Equation 7 is common
within the literature, this methodology can produce larger variability in iron solubility across grid
cells because it is based on both soluble and total iron annual mean concentrations. In the online
method, variability is reduced as extreme values in soluble and total iron concentrations generally
do not occur at the same time. We can define the occurrence of extreme values, with respect to
the time frame considered, by analysing a relative Z-score metric, calculated as follows:

$$Z_{Fe,t} = \frac{(Fe_t - \overline{Fe_t})}{\sigma Fe_t} \qquad or \qquad Z_{Fe,s} = \frac{(Fe_s - \overline{Fe_s})}{\sigma Fe_s} \qquad \text{Equation 8}$$


where Fe is either total ($Fe_t$) or soluble ($Fe_s$) iron. The relative normalized Z-score can then be
calculated as follows:

$$\sum_{i=1}^{n} (z_{t,i} - z_{s,i}) / z_{t,i} \qquad \text{Equation 9}$$


where $Z_{t,i}$ and $Z_{s,i}$ are the Z-scores of total and soluble iron concentrations, respectively, at each
grid cell for each time step *i*. The Z-score metric provides a relative direction and distance of an
instantaneous value with respect to its mean. The Z-score is reported in multiples of the standard
deviation (Equation 8); therefore, a Z-score of zero indicates that the data point value is identical
to the mean value. To assess the relative difference in the variability, at a given time, between
the modelled total and soluble iron concentration and its mean we calculated the difference in Z-
scores between total and soluble iron concentrations and normalized it using the Z-score of total
iron concentration (Equation 9). Note that the Z-score of the soluble iron concentration could also
be used to normalise the difference. This method allows for the examination of how the
occurrence of extreme concentration values in total and soluble iron influences the method of
solubility calculation (Equation 6 vs. Equation 7).

**2.6 Iron ocean deposition source apportionment**
An ocean deposition source apportionment sub-study was designed to classify ocean deposition
regions according to the dominant atmospheric soluble iron source, rather than ocean basins
defined from a more traditional physical oceanographic viewpoint (e.g., Gregg et al., 2003). By
incorporating recent model estimates for dust and the importance of pyrogenic iron emissions
(Luo et al., 2008; Matsui et al., 2018) the seven large-scale source regions defined in Mahowald
et al. (2008) were modified slightly to separate the major dust iron source regions from fire and
anthropogenic combustion iron source regions. This resulted in a total of 10 iron emission source
regions (Fig. 1; see also Table S1 for details).
Simulations in the source apportionment study used BAM-Fe, as described in Scanza et al. (2018)
with slight modification. Briefly, anthropogenic combustion iron emissions were increased by a
uniform factor of five, and iron from fires followed the updated Fe:BC ratio (Table 4) and seasonal
variability in the fire BC emissions; all as per MIMI. Aerosols were externally mixed in BAM, and
therefore altering the regional aerosol loading did not affect aerosol transport or deposition in the
more significant way it could in MAM, in which aerosol are internally mixed. This information was
then used in Section 4.3.1 to compare the differences in daily mean deposition of soluble iron
between the BAM-Fe and MIMI models within each defined ocean region.

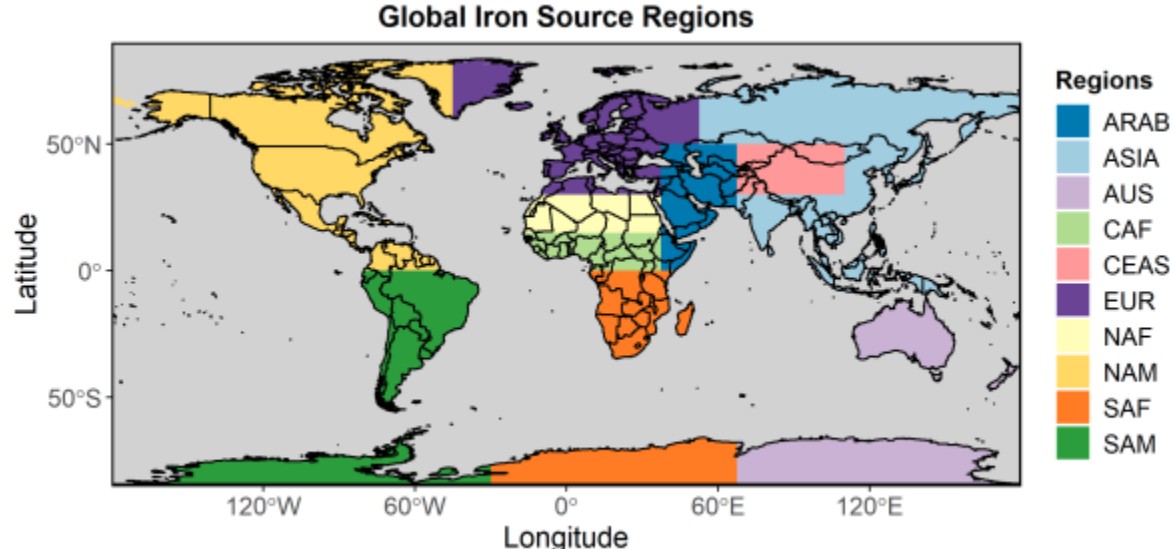


**Figure 1**. Major iron aerosol emission source regions.

**3 Modelled dust and iron aerosol concentrations compared to observations**
In terms of Earth system modelling, and the biogeochemistry that connects the land–atmosphere–
ocean components, we are ultimately motivated here to improve the magnitude of the atmosphere
to ocean iron deposition flux and its fractional solubility (from which the soluble iron flux can be
derived). We compare the model results with a series of observations, and herein, highlight some
of the problems discovered when directly comparing with a sporadic (in both space and time)
observation dataset, as is currently common practice (Myriokefalitakis et al., 2018).

**3.1 Global dust comparisons**
Comparison of dust AOD with regional dust AOD observations (Fig. 2) from the AERONET
observational datasets (Holben et al., 2000), as subsampled in Albani et al. (2014), shows good
agreement globally (correlation: $r^2 = 0.64$). This results in MAM annual global mean emissions of
3250 ± 77 Tg dust $a^{-1}$ (Aiken = 16 Tg $a^{-1}$, accumulation = 36 Tg $a^{-1}$, coarse = 3198 Tg $a^{-1}$), which
is at the higher end of literature estimates of ~500–4000 Tg dust $a^{-1}$ (Bullard et al., 2016; Huneeus
et al., 2011; Kok et al., 2017). Dust emissions in MAM are 84 ± 4% higher than our previous mean
of 1768 Tg dust $a^{-1}$ in BAM (Scanza et al., 2018), because dust lifetime has proportionally
decreased (Table S2) which affects coarse mode dust aerosol (where 98 – 99% of total dust mass
is emitted) more than fine mode dust aerosol. Globally, both dust concentrations (correlation: $r^2 =$
0.89) and deposition (correlation: $r^2 = 0.83$) are simulated well compared to observation within
MIMI. A higher correlation of modelled dust concentrations with observations is calculated in the
Northern Hemisphere (NH; $r^2 = 0.89$) compared to the Southern Hemisphere (SH; $r^2 = 0.67$), but
with gradient of line of best fit is further from 1:1 (NH: 1.22 vs. SH: 1.07). Conversely, for dust
deposition a lower correlation with observations is simulated in NH ($r^2 = 0.75$) compared to the
SH ($r^2 = 0.60$) but with a gradient of the line of best fit closer to 1:1 (NH: 1.07 vs. SH: 0.72). Overall,
results presented in this study suggest an improvement  on previous dust modelling complications
related to underestimating dust deposition when tuned to dust concentration (Huneeus et al.,

503    2011).

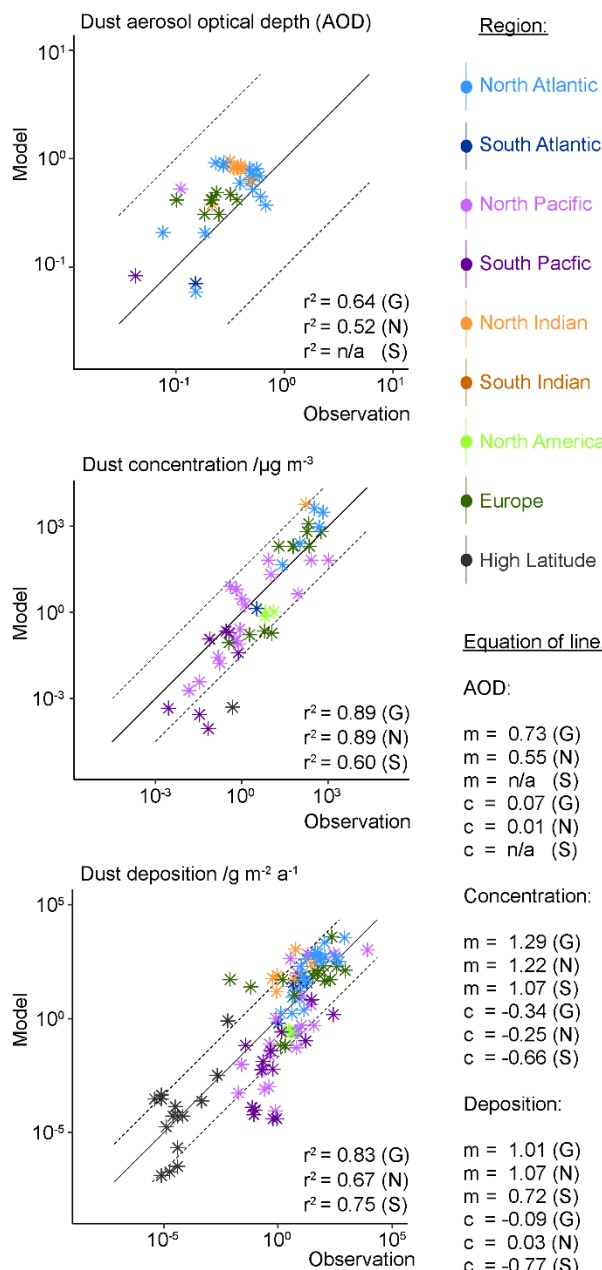


**Figure 2.** Dust aerosol optical depth, surface concentrations and deposition in modal aerosol model and observations (Albani et al., 2014; Holben et al., 2000). Correlation ($r^2$), gradient (m) and intercept (c) shown for global (G), Northern Hemisphere (N) and Southern Hemisphere (S) regions.

509

**3.2 High latitude dust and iron aerosol**

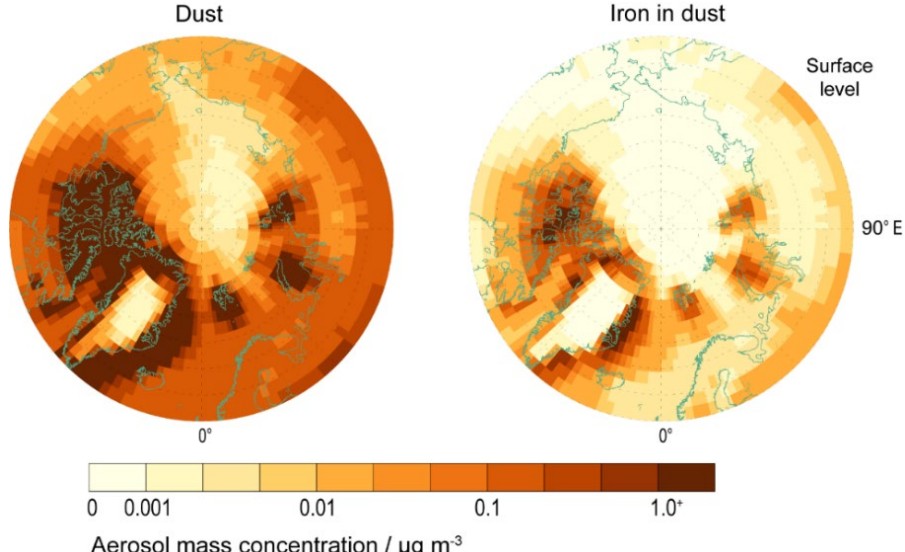

**Figure 3.** High latitude (>60ºN) dust (sum of eight mineral species and four dust-iron species) and iron (sum of four dust-iron species) mass concentrations (µg m$^{-3}$) at the surface model level.

Including the parametrization of Kok et al. (2014a) removes the requirement of a soil erodibility map (Table 1). In addition, in previous versions of the model, the high latitude dust sources were zeroed, because there were no observations at that time for high latitude sources of dust (Albani et al., 2014). However, more recent observations have suggested high latitude dust sources do exist (Bullard et al., 2016; Crusius et al., 2011; Tobo et al., 2019), often related glacial processes (Bullard, 2017) with a higher fraction of bioavailable iron relative to lower latitude dust sources (Shoenfelt et al., 2017). Thus, for the new version of the model we have allowed for the inclusion of high latitude dust sources (Fig. 3). In general, aerosol dust and iron concentrations peak closest towards the coast lines and during summer. Emissions of dust from >50ºN are ~1.3 ± 0.2% of the global dust total, which is half of the estimates derived from field and satellite data at 2–3% of the global total (Bullard, 2017; Bullard et al., 2016). However, the resulting magnitude and seasonality of dust concentrations has been shown in a recent study to be consistent with observed measurements from Svalbard (Tobo et al., 2019).

     **3.3 Global iron aerosol concentration and fractional solubility**

532

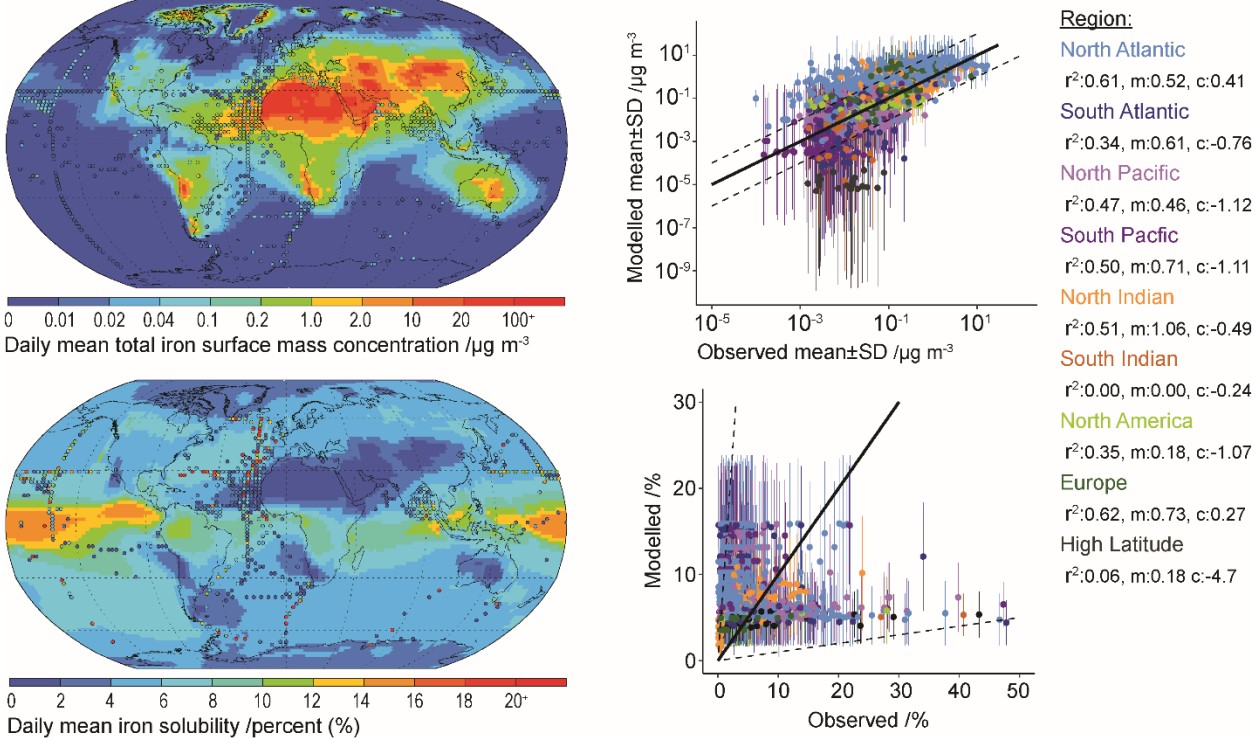

533

534

**Figure 4.** Daily mean model total iron concentration and solubility from 2007 to 2011. Observations (circles) overlaid (at resolution of the model grid) as a mean from 1524 individual records in Mahowald et al. (2009) and in Myriokefalitakis et al. (2018). Also shown are scatter plots of the model mean and standard deviation compared to each available observation and identified by oceanic region. Correlation ($r^2$), gradient (m) and intercept (c) for total iron with observations shown for each region.

541

542

There are several propositions explaining the sources of soluble iron, and the inverse relationship between total iron amount and iron solubility (Sholkovitz et al., 2012). While total iron mass concentrations are dominated by desert dust sources, soluble iron can be a product of mineral dust processed in the atmosphere or emitted from pyrogenic sources (Chuang et al., 2005; Guieu et al., 2005; Ito et al., 2019; Luo et al., 2008; Meskhidze et al., 2003; Schroth et al., 2009). Previous

studies have shown that either of these can explain the inverse relationship, and that the spatial
distribution of data is required to provide more information (Mahowald et al., 2018). Therefore, we
explored how to best use the spatial data to compare with the model results. The five-year (2007
to 2011) mean iron concentration from MIMI is compared to an extensive dataset of observations
of total iron and its fractional solubility (Fig. 4). The model captures the global mean observational
total iron concentration well; however, relatively low regional correlations ($r^2 < 0.4$) occur in the
South Indian ($r^2 = 0.0$), South Atlantic ($r^2 = 0.34$), North America ($r^2 = 0.35$) and high latitude ($r^2 =$
$0.06$) ocean regions, suggesting future model improvements can be focused here.
In the absence of iron atmospheric process modelling, ocean biogeochemistry models with an
iron component (e.g., Aumont et al., 2015; Moore et al., 2004) have estimated iron solubility from
offline dust modelling by means of an assumption that it contains 3.5% iron by weight, of which
2% is soluble. Iron solubility is highly temporally and spatially variable however, and in the
absence of spatial atmospheric emission information, pyrogenic iron sources, and atmospheric
processing of iron an estimate of 2% solubility leads to underestimates of observed iron solubility
in nearly all HNLC ocean regions (Fig. 4).
Aggregating observations onto a lower resolution grid (sometimes termed 'super-obbing')
compared with the model can help reduce the representation error when comparing with such
limited observations (Schutgens et al., 2017). Fig. 5 uses an observational resolution one-third
that of the model and the model-to-observation comparison of the mean state is thus improved.
Persistent observation-based features of the local environment become more obvious while,
conversely, less frequent ones diminish. At this observational resolution, the low total iron
concentrations in the North Atlantic ~30ºN, as seen in Fig. 4, are perhaps not a common feature,
and the model much more precisely represents the climatological state here than Fig. 4 might
suggest. However, examining the North Pacific reveals that the model imprecisely represents the
mean state here. Potential missing iron sources in remote regions, such as the North Pacific,
include: (1) shipping emissions (Ito, 2013), which have a high soluble iron content from oil
combustion (Schroth et al., 2009); (2) volcanic emissions, which provide a localized "fertilizer" to
the surface ocean owing to the macronutrients and trace metal nutrients contained within them
(Achterberg et al., 2013; Langmann et al., 2010; Rogan et al., 2016); and (3) low Asian and South
American aerosol concentrations, either through underrepresenting combustion emission sources
(Matsui et al., 2018) or in the transport and deposition of aerosol within these regions (Wu et al.,
2018). These are discussed in more detail in the discussion Sections 5.1 and 5.2.


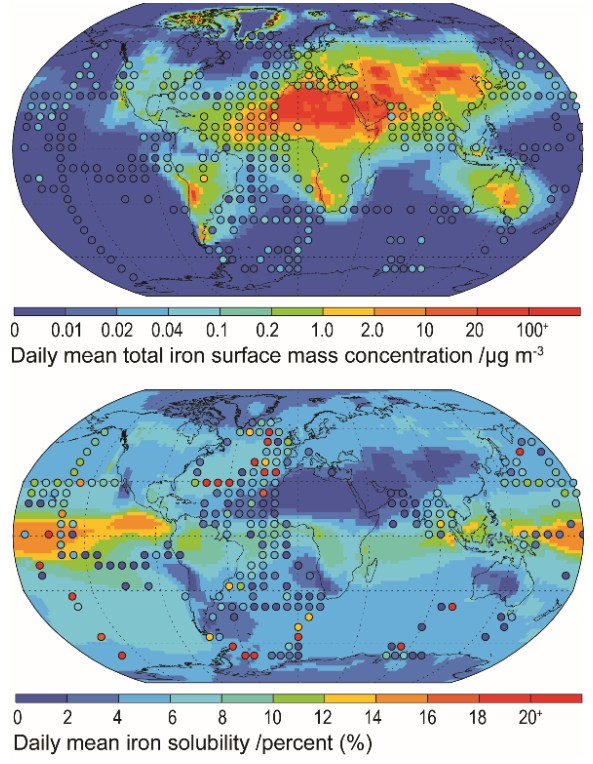



**Figure 5.** Daily mean model total iron concentration and solubility from 2007 to 2011. Observations (circles) overlaid (at resolution one-third of the model grid) as a mean from 1524 individual records in Mahowald et al. (2009) and in Myriokefalitakis et al. (2018).



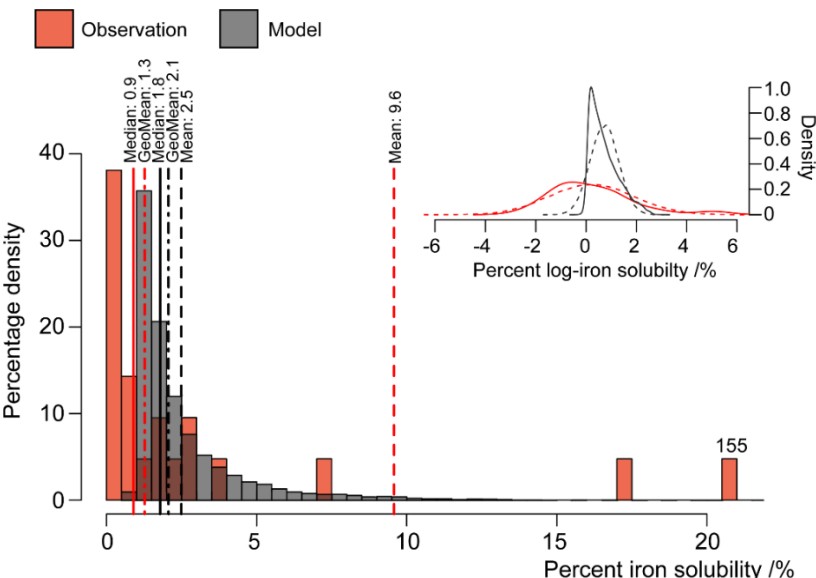

**Figure 6.** Histogram of observations (n = 21) and daily model results (2007 to 2011) of iron solubility between 16 to 20⁰N and 27 to 32⁰W (one observation point and nine co-located model grid cells in Fig. 4). Mean (dashed lines), geometric mean (dot-dash lines) and median (full line) values shown above respective dataset colour line. Note that the single observation value of 155% is off the scale and placed as such with value given above. **Insert.** Log-plot for the same data (solid lines) with projected log-normal distribution from mean and standard deviation of data (dashed lines).

In terms of iron solubility (soluble iron concentration / total iron concentration), the model is not capturing the observational mean state in many regions (Fig. 5). A detailed examination of the observation point at 18⁰N and 330⁰E (anomalous green point surrounded by blue points in the North African outflow plume in Fig. 4) and the nine model grid cells co-located with it in Fig. 6 shows how a single high observation (155% percent solubility) is causing a representation issue (see also section 4.3.1 regarding soluble iron deposition). Both model and observation histogram distributions are similar, as are the median (model: 1.8, observation: 0.9) and geometric mean (model: 2.1, observation: 1.3) values. However, the arithmetic means are not similar (model: 2.5, observation: 9.6) and while a high observation value of 155% is likely to be an outlier, and should be at most 100%, it still informs us about what is possible and simply discounting it (even at an adjusted 100%) would require strong justification. It is therefore advisable to instead alter the estimator of the average. Comparing model to observation differences calculated using the median or geometric mean reveals that they are similar in magnitude, as one would expect for

log-normally distributed data (Fig. 6 insert). Although the median is robust with respect to outliers,
the model results may not exhibit a uniform Gaussian distribution (Fig. 6 insert; solid compared
to dashed lines) and often the amount of available observations is also low (Fig. 7) suggesting
that its use also requires careful consideration. An equivalent methodology to the geometric mean
in Fig. 7 would be to first log transform the data before calculating the arithmetic mean. Arguments
pertaining to the appropriate methodology for comparing model results to temporally limited
observations extend beyond the iron aerosol examination in this study to all aerosol comparisons
with limited observations.

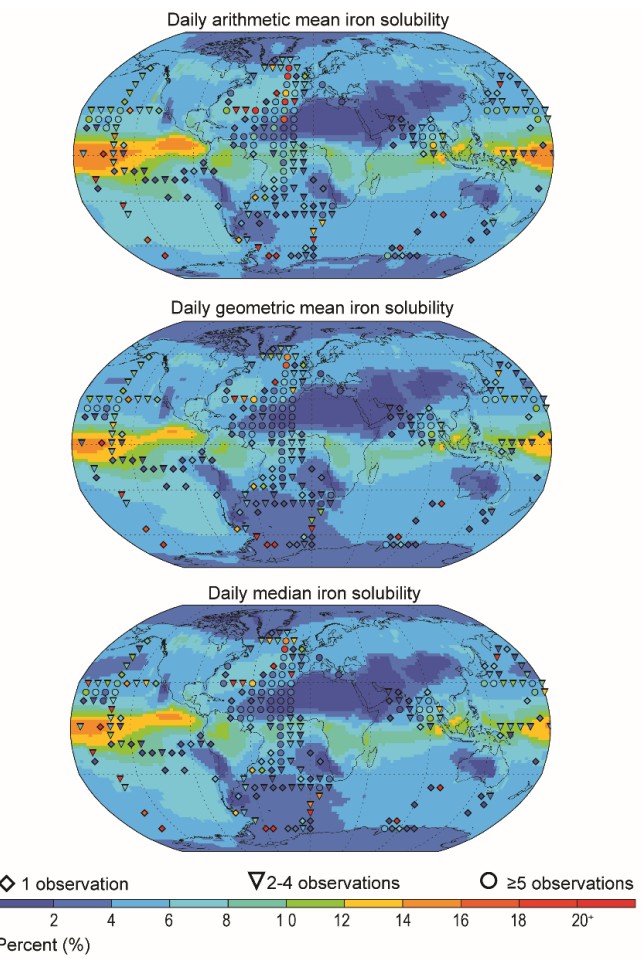


**Figure 7.** Daily arithmetic mean, geometric mean, and median model solubility (2007 to 2011).
Observations overlaid (at resolution one-third of the model grid) as either the arithmetic mean,
geometric mean or median, respective to the model averaging. Number of observations denoted
by symbol: lowest confidence (one observation, diamond); intermediate confidence (two to four
observations, triangle); highest confidence (five or more observations, circle).

### 3.4 Calculating iron solubility

It is interesting to note the effect that the order of operations (taking the average-of-ratios compared to the ratio-of-averages) has when calculating iron solubility (Fig. 8). Throughout this study, percent iron solubility was calculated at each model time step (30 minutes) and then the daily mean output analysed (online; Equation 6) at an annual or 5 year mean time resolution. It is also acceptable to use the simulated soluble and total iron concentrations to generate the annual or 5 year mean iron solubility in a postprocessing step (offline; Equation 7). The resulting differences between methods are not insignificant however, with the offline method creating a distribution in which low iron solubility is generally lower and the highest (>18%) iron solubilities are generally higher. Overall, global annual mean iron solubility calculated online is one-third (34%; NH=40%, SH=29%) higher than when calculated offline.

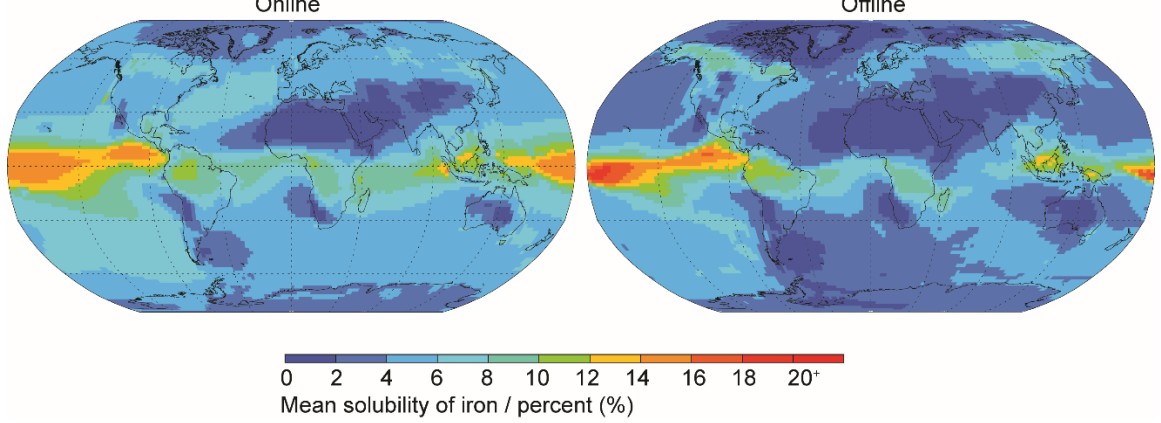

**Figure 8**. Mean solubility of iron when solubility is calculated at each 30 min model time step ('online') and when it is calculated post processing from the daily mean soluble and total iron concentration ('offline').

The average relative Z-score (Equations 8 and 9) is around zero for most model grid cells (Fig. 9) indicating that they mostly followed similar temporal and relative magnitude trends. However, even if the average relative Z-scores are around zero and the ratio of relative standard deviations is around one, the ratio of online:offline calculated iron solubility is most likely >1. Temporal differences in the soluble and total iron concentration might therefore be controlling the overall

solubility at each model grid cell. We also find that the ratio of online and offline solubility is >1 for
most of the cases when the ratio of relative standard deviations of soluble and total iron is <1 (Fig.
S2), indicating that the differences in both methods of iron solubility calculation are sensitive to
the differences in relative size of the tails of the distribution. That is, if soluble iron has narrower
tails compared to total iron at any grid cell, it is highly likely that a higher solubility will be obtained
in the online method compared to the offline.. The extreme ratio of the tails of soluble and total
iron are only found in specific regions with highest temporal variability in emissions and modelled
solubilization of insoluble iron (Fig S2).

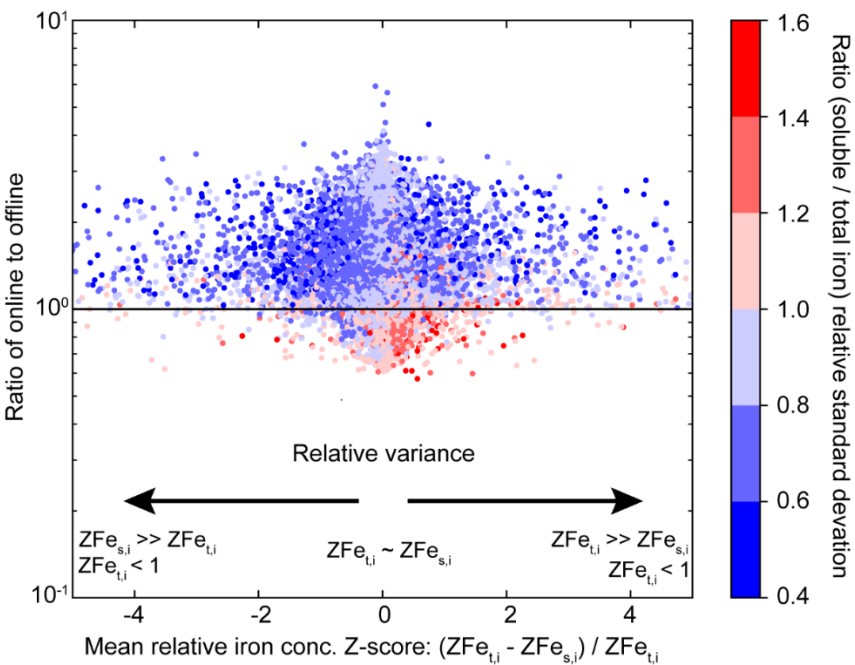


**Figure 9.** Relationship of online to offline derived iron solubility to the relative Z-score for total
(ZFe$_t$) and soluble (ZFe$_s$) iron and the relative standard deviation (σFe / $\overline{Fe}$ ) at each grid cell for
the year 2007.

Field measurements have generally suggested an inverse relationship between total and soluble
iron concentrations (Myriokefalitakis et al., 2018). This means that high total iron concentrations
are generally accompanied by low soluble iron concentrations and vice versa. By assuming that
the field measurements faithfully represented the actual average values of soluble and total iron
concentration at those locations, we implicitly assume that all the measurements have a Z-score
of zero. In Fig. 9 we show that this is not the case with the modelled results, and the two variables
can be relatively farther from their respective means even when averaged over the modelled time
period.
Sensitivity of a result to the order of operations extends beyond iron solubility to any variable that
is calculated in a similar manner, and current multi-model inter-comparison project (MIP) protocols
do not explicitly account for this. However, the effects of outliers, in both online and offline
methods, can be reduced by employing the geometric mean and has been used in some MIP's
(e.g., Mann et al., 2014). It will be also be important to consider differences in the solubility of iron
induced by the choice of the order of operations as ocean biogeochemical models move away
from using offline results from global climate or chemistry transport models to online results within
Earth system models, which are designed to couple the two components at each time step. For
short term interactions between deposited iron and ocean biota shorter term averaging may be
more important (e.g., Guieu et al., 2014), but for long term period accumulation of iron that is
(re-)cycling in the oceans, the longer term average may be more appropriate (Moore et al., 2013).
One should be aware, however, that iron is readily removed from the ocean mixed layer, and
thus, the lifetime of iron may well be short enough for the 'online' calculation to be more
appropriate much of the time (Guieu et al., 2014).


**4.0 MIMI vs. BAM-Fe**
In this section, we discuss how the new modal aerosol mode version of MIMI compares to its
predecessor bulk aerosol model version (BAM-Fe) throughout all three stages of the atmospheric
iron life-cycle.


**4.1 Iron emission comparison**
Globally averaged emissions of dust (3200 Tg a$^{-1}$) and its iron component (126 Tg a$^{-1}$) are within
the current multi-model range (Table 7). The simulated annual mean iron in dust percentage is
4.1%, with the highest percent occurring in the coarse mode at 6.5% and lowest percent occurring
in the Aiken mode at 1.1%. Accounting for dust minerology therefore increases the global mean
iron percent by weight above the currently well-used global mean estimate of 3.5% (e.g., Jickells
et al., 2005; Shi et al., 2012).

**Table 7.** Dust, fire, and combustion emissions of iron and relevant co-emitted aerosol emissions
(to two significant figures). Multi-model emission range from the four global atmospheric iron
models (including BAM-Fe) reported in Myriokefalitakis et al. (2018). Fine (sum of Aiken and
accumulation modes) and coarse (coarse mode) size mass emissions also given for dust, fire iron
and combustion iron.

| | Annual mean emissions /Tg a$^{-1}$ | | | |
|---|---|---|---|---|
| | BAM-Fe | MIMI | Luo et al. (2008) | Multi model |
| Dust | 1800 | 3200 | 1600 | 1200–5100 |
| Fine, Coarse | 20,1700 | 50, 3200 | | |
| | | | | |
| Dust iron | 57 | 130 | 55 | 38–130 |
| Pyrogenic iron (Fire&Comb.) | 1.9 | 5.5 | 1.7 | 1.8–2.7 |
| | | | | |
| Fire BC | 4.1 | 2.6 | 3.6 | |
| Total fire iron | 1.2 | 2.2 | 1.1 | |
| Fine, Coarse | 0.08, 1.1 | 0.30, 1.90 | 0.07, 1.00 | |
| | | | | |
| Combustion BC | 4.6 | 5.0 | 5.0 | |
| Total comb. iron | 0.66 | 3.3 | 0.66 | |
| Fine, Coarse | 0.10, 0.56 | 0.50, 2.80 | 0.10, 0.56 | |


Compared to BAM-Fe, MIMI dust emissions are ~80% higher and the iron it contains is ~120%
higher(Table 7). Although both the BAM-Fe and MIMI models are globally tuned to a similar dust
AOD (~0.03), and based within the same host model (CESM), changing from a bulk aerosol
scheme (e.g., Albani et al., 2014; Scanza et al., 2015) to a modal aerosol scheme  reduces the
aerosol lifetime significantly (Liu et al., 2012 and Table S2). The spatial distribution of dust
emissions is also different following the move to the Kok et al. (2014a, 2014b) parameterization
(Table 1), resulting in the spatial distribution of dust AOD also altering (Fig. S3). Total pyrogenic
iron emissions (sum of fires and anthropogenic combustion activity) in MIMI are higher than
previous estimates by a factor of between two and three (Table 7), reflecting the recently growing
evidence indicating that they have been previously underestimated (Conway et al., 2019; Ito et
al., 2019; Matsui et al., 2018).

**4.2 Iron atmospheric processing comparison**
There is a much lower aerosol pH in the fine aerosol modes (Aiken and accumulation) in MIMI
compared to that in BAM-Fe (Fig. 10). This is due to a combination of resolving pH in each aerosol
size mode in MIMI and the subsequent lowering of the pH value (1) being applied in the two fine
aerosol modes (Aitken and accumulation). Conversely, dust dominating the coarse aerosol mode
provides more of an opportunity for [Calcite] > [$SO_4$] in this aerosol size fraction, resulting in most
continental areas having a high coarse mode aerosol pH in MIMI compared with the higher pH
being much more localized to the major desert regions in BAM-Fe. Acidic processing of iron in
MIMI therefore proceeds faster globally in the fine sized aerosol modes (Aitken and accumulation)
compared to the BAM-Fe fine size bin (0.1-1μm), but generally slower over continental regions in
the coarse mode than in BAM-Fe coarse size bins (1-10 μm).
Comparison of Fig.10 to modelled pH estimates by Myriokefalitakis et al. (2015) shows generally
good agreement in the NH, but in the SH MIMI simulates less acidic coarse mode aerosol over
continental regions and more acidic aerosol over marine regions. As iron models are unable to
capture the high observed iron solubility (>10%) over SH marine regions (Myriokefalitakis et al.,
2018), and in the absence of remote pH aerosol observations, we suggest that our basic
parameterization captures an aerosol pH which is suitable for use in Earth system models

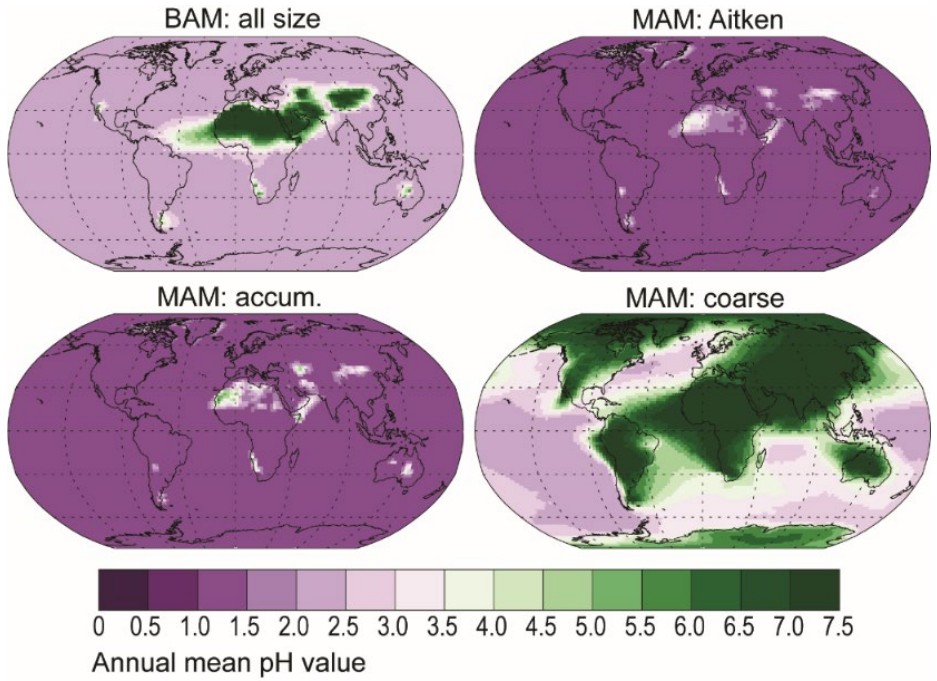

Annual mean pH value

**Figure 10**. Surface level annual mean interstitial aerosol pH. If [SO4] > [Calcite] then pH = 1 in Aitken and accumulation modes or 2 in coarse, else pH = 7.5 (Table 5).

Model physics, and hence simulated cloud cover, is significantly different between CAM4 and CAM5. Fig. 11a shows the relative model difference in the oxalate distribution between MIMI, which also includes an increase in the tuning factor by an order of magnitude (from 15 to 150; Table 5), and BAM-Fe by normalising by the simulated cloud fraction in each model respectively. The effect of oxalate on iron dissolution is therefore larger in MIMI over extra-tropical ocean regions, where iron models underrepresent solubility (Myriokefalitakis et al., 2018), and land regions which are dense in tropical vegetation or industry (both centres of large aerosol precursor gas emissions). Compared to observations (Myriokefalitakis et al., 2011; Table S3) modelled oxalate concentrations are well represented at high observed concentrations but are biased low when observed concentrations are low (Fig. 11b). The low model bias is stronger within remote observational regions (marine vs. urban observation sites), suggesting that the removal of secondary organic aerosol may be too strong within the model and/or that there is a missing marine aerosol pre-cursor gas emissions source (Facchini et al., 2008; O'Dowd and de Leeuw, 2007) in this model which significantly lowers simulated secondary organic aerosol, and thus oxalate, concentrations.

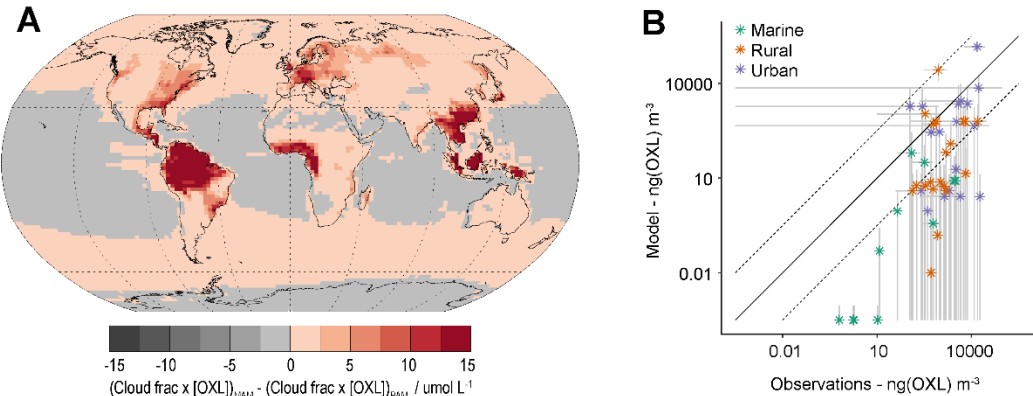

**Figure 11. A:** Relative difference in organic ligand reaction on in-cloud iron aerosol dissolution between MIMI and BAM-Fe. Due to significant differences in simulated cloud cover between CAM4 and CAM5 oxalate concentrations [OXL] are multiplied by the model simulated cloud fraction in this figure. **B:** Surface level oxalate (OXL) concentration in the model and observations. Model values are annual mean (2007-2011) and monthly standard deviation. Observation values are from Table S3 in Myriokefalitakis et al. (2011) and reported with uncertainty where given.

Comparison of mineral dust and pyrogenic sources of modelled soluble iron (sum of emissions and atmospheric dissolution; Fig. 12) with the four iron models (including BAM-Fe) reported by Myriokefalitakis et al. (2018) shows that the spatial distribution in MIMI is broadly similar for most regions of the world. A notable difference exists in the North Pacific region where the soluble iron source in MIMI is lower than all other iron models, and similarly with total iron concentrations when compared to observations (Figs. 4 and 5). Future development of MIMI should thus be focused on the North Pacific, including the addition of shipping soluble iron emissions which are relatively concentrated in this region (Ito, 2013). An improvement for MIMI can be seen over the Atlantic region directly downwind of Saharan soluble iron sources. In general, iron models are over representing iron solubility close to dust sources compared to observations (Myriokefalitakis et al., 2018) and in order for BAM-Fe to reach better agreement with observed iron solubility in this region dust emissions of soluble iron had to be scaled downwards (Conway et al., 2019). We suggest this improvement is linked to the improved modal representation of aerosol pH in MIMI (Fig. 10).

778

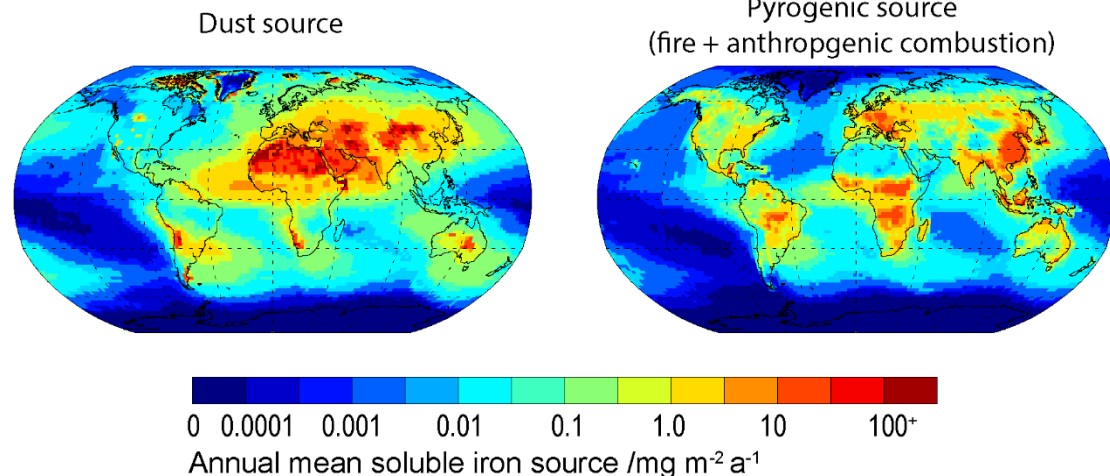

0 0.0001 0.001 0.01 0.1 1.0 10 100⁺
Annual mean soluble iron source /mg m$^{-2}$ a$^{-1}$


**Figure 12.** Annual mean dust and pyrogenic (sum of fires and anthropogenic combustion)
soluble iron source (i.e., sum of emissions and atmospheric processing).

**4.3 Iron ocean deposition flux comparison**
Similar to the previous study by Scanza et al. (2018), we report the amount of total and soluble
iron deposited in each of the major ocean basins (Table 8) as defined by Gregg et al. (2003). We
find that, in MIMI the amount of total iron deposited to all ocean basins is approximately double
that estimated in BAM-Fe (26 vs. 12 Tg Fe a$^{-1}$, respectively), while soluble iron deposition is similar
(~0.5 Tg Fe a$^{-1}$ in both models). The larger mineral dust emission flux in MIMI (3200 Tg dust a$^{-1}$
compared to BAM-Fe dust emission of 1800 Tg dust a$^{-1}$) is driving most of the increases to total
iron deposition because it is the primary iron source (Table 7). In general, the magnitude of soluble
iron deposition to the oceans is more evenly distributed across hemispheres in MIMI owing to a
major reduction (approximately one half) in the equatorial North Central Atlantic basin deposition
flux and increases to SH ocean deposition fluxes of a factor of two to four. In MAM4 dust is treated
as internally mixed aerosol with sea salt, leading to higher rates of wet deposition than when dust
is externally mixed aerosol (Liu et al., 2012) as it is in CAM4. The internally mixed treatment of
dust aerosol in MAM4 is thus an important factor leading to the lower simulated dust lifetime when
compared to BAM-Fe (Table S2). Over the North Central Atlantic region, the combination of a
lower soluble iron source (Fig. 12 compared to Fig S4b by Myriokefalitakis et al. (2018)), dust
atmospheric lifetime (Table S2), lower aerosol pH (Fig. 10), and lower relative organic ligand
processing (Fig. 11) will all work towards reducing the magnitude of atmospheric soluble iron

deposition flux in MAM4 compared to BAM-Fe. There are significant increases in anthropogenic combustion iron deposition in all equatorial and NH ocean basins, driven by the 5-fold increase in combustion emissions implemented in MIMI. The percent contribution from pyrogenic iron to total iron deposition between MIMI and BAM-Fe is however more similar for all northern and equatorial oceanic regions than southern oceanic regions. Beyond the correction to anthropogenic combustion emissions, which are NH dominated, this could be due to differences in the emissions of both dust and fire aerosol, structural differences between models relating to the aerosol size and composition which alters aerosol deposition rates, or a lower soluble iron source (Fig. 12); it is most likely to be a combination of all three.

**Table 8.** Global and regional ocean basin deposition (Gg a$^{-1}$) of total and soluble iron in BAM-Fe (Scanza et al., 2018) and MIMI (this study). Deposition was multiplied by the ocean fraction of model grid cell and is reported at two significant figures. Percent contribution from pyrogenic (sum of fires and anthropogenic combustion) iron sources to deposition also given. Ocean basins are those defined by Gregg et al. (2003) and previously used by Scanza et al. (2018).

| | Dust and comb. deposition /Gg a$^{-1}$ | | | | Percent iron from pyrogenic sources /% | | | |
| | Total iron | | Soluble iron | | Total iron | | Soluble iron | |
| | BAM-Fe | MIMI | BAM-Fe | MIMI | BAM-Fe | MIMI | BAM-Fe | MIMI |
|---|---|---|---|---|---|---|---|---|
| **Global** | **12000** | **26000** | **500** | **530** | **3.3** | **5.0** | **7.6** | **23** |
| N. Atlantic | 1800 | 5300 | 46 | 86 | 1.9 | 2.9 | 4.8 | 11 |
| N. Pacific | 730 | 1200 | 35 | 36 | 10 | 19 | 15 | 43 |
| NC. Atlantic | 2900 | 5700 | 92 | 89 | 0.30 | 0.52 | 0.9 | 3.7 |
| NC. Pacific | 230 | 300 | 16 | 12 | 7.9 | 24 | 10 | 56 |
| N. Indian | 2700 | 7000 | 62 | 101 | 1.2 | 2.1 | 3.9 | 10 |
| Eq. Atlantic | 2600 | 2600 | 190 | 95 | 2.8 | 9.9 | 5.5 | 34 |
| Eq. Pacific | 59 | 91 | 6.2 | 6.7 | 21 | 37 | 25 | 68 |
| Eq. Indian | 830 | 1200 | 35 | 39 | 5.9 | 12 | 11 | 38 |
| S. Atlantic | 65 | 790 | 4.1 | 16 | 30 | 4.8 | 50 | 25 |
| S. Pacific | 21 | 250 | 1.4 | 6.4 | 41 | 7.8 | 50 | 30 |
| S. Indian | 42 | 200 | 3.0 | 6.9 | 51 | 16 | 58 | 46 |
| Antarctic | 270 | 1300 | 12 | 37 | 20 | 12 | 48 | 44 |

The fraction of fire aerosol which is injected above the boundary layer is crucial for determining
its capacity for long range transport (e.g., Turquety et al., 2007). Vertically distributing fire iron
emissions in MIMI, as compared to emitting all iron from fires at the surface as in BAM-Fe,
increases the long-range transport of iron aerosol to remote ocean regions (Fig. 13). In general,
vertically distributing fire emissions results in small increases in soluble iron deposition (between
0 and 20%) in SH ocean regions and a larger increase (between 20 and 40%) to NH oceans, with
converse lower land deposition close to the major regions of fire activity. The exception being in
the sub-Arctic North Pacific, a HNLC region, where iron deposition from fires significantly
increased until more than doubling that when surface fire emissions are used.
The dry deposition flux is sensitive to the aerosol properties, surface roughness and modelled
turbulence. Although increasing the vertical resolution has been shown to increase surface $PM_{10}$
concentration (Menut et al., 2013) and better simulate  the dust vertical profile (Teixeira et al.,
2016), it is not as yet clear if this would correspondingly increase the dry deposition flux.

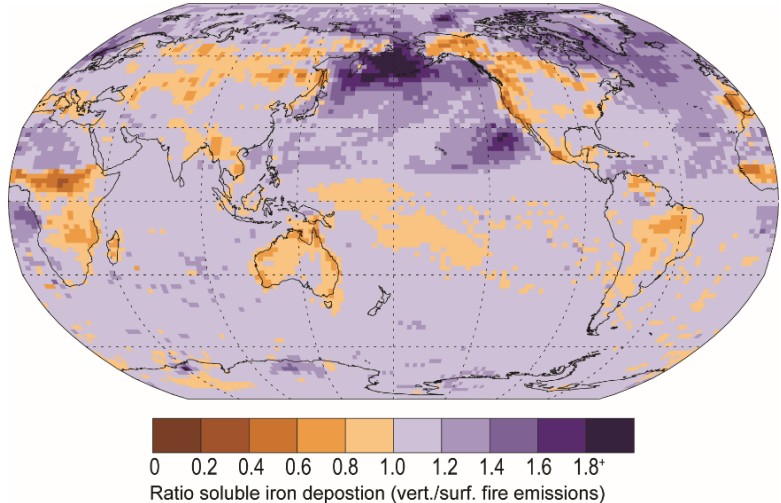

Ratio soluble iron depostion (vert./surf. fire emissions)



**Figure 13.** The ratio of soluble iron deposition from fires when emissions are emitted with a
vertical distribution to fires compared with when emission are only at the surface (i.e.,
vertical/surface). Single year (2007) comparison only.

**4.3.1 Source region comparison**
The eight regions in Fig. 14 are chosen based on 10 (one for each region in Fig. 1) simulations
undertaken using the modified version of BAM-Fe described in the methods Section 2.6. The
emission region (Fig. 1) with the highest fractional contribution to the total soluble deposition flux
in each grid cell was examined and from this the boundaries of each region in Fig. 14 delineated.
The resulting eight ocean iron deposition regions are split equally into four in the NH and four in
the SH. Note, however, that the NH–SH divide sits at 15º S, and not the equator, which is due to
transport differences in each hemisphere and the position of the Inter Tropical Convergence Zone.
Of the four regions that can be defined as being major dust deposition receptors (Fig. 14; bottom
panel bar chart) the North Indian Ocean (#1), North Atlantic and Central Pacific (#4), and South
America dust (#7) regions have a single dominant source each, while the North Pacific (#3) region
is more variable. These dust-dominated iron deposition regions are similarly reproduced by other
global iron models (Ito et al., 2019; Myriokefalitakis et al., 2018). The regions of the Southern
Hemisphere Oceans (#5) and Australian and South Pacific (#6) receive similar amounts of mineral
dust and pyrogenic iron, suggesting that the iron sources are spatially closer and, thus, share
much more similar transport pathways than the South East Asian Ocean (#2) and South America
Pyrogenic (#8) regions, which have a much more distinct pyrogenic iron source signal. Deposition
regions are more clearly defined when using this methodology compared to those from a more
traditional classification of ocean basins based on physio-geographical oceanography (Fig. S4).
This information can be used to assess which ocean regions are most likely to be affected by
anthropogenic perturbations to the magnitude of iron sources within different regions, whether
through land use land cover change or industrialization.

The variability in the daily soluble iron deposition flux to each of the eight ocean regions, as seen
in Fig. 14, is much larger in MIMI than it is in BAM-Fe (Fig. 15), reaching over 10 orders of
magnitude between the minimum and maximum flux in many regions. This is due in part to the
increased variability in fire emissions, which was improved in MIMI to track the BC emitted from
fires, and switching from the offline soil erodibility map used in BAM-Fe to the Kok et al. (2014a)
physical based emission parametrization used in MIMI. Anthropogenic combustion emissions are
temporally static in both model frameworks, and therefore do not affect the variability in this study
as much as fires and mineral dust but will in future if this is changed to represent a seasonal
emission cycle. We can see that each of the dust and fire updates in MIMI are having a large
impact by comparing the Patagonian dust dominated South America Dust (SADU) region and the
fire dominated South America Pyrogenic (SAPY) region. Most of the dust deposited (30 to 90%)
in the ocean occurs during large dust events that are on just 5% of the days (Mahowald et al.,
2009) resulting in large differences between median and mean deposition amounts in all regions,
as seen in Fig. 15. It is important to note that the mean is always above the inter-quartile range,
further supporting our previous arguments pertaining to the modelled mean not being an ideal
estimate of the average as it does not represent the log normal distribution of aerosol. Comparing
the mean:median ratio suggests that extreme dust events are also more pronounced in MIMI
(CAM5) than those in BAM-Fe (CAM4).

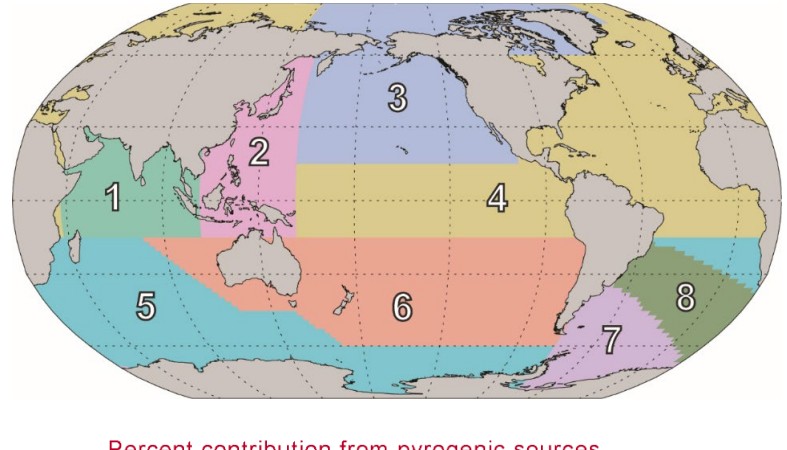


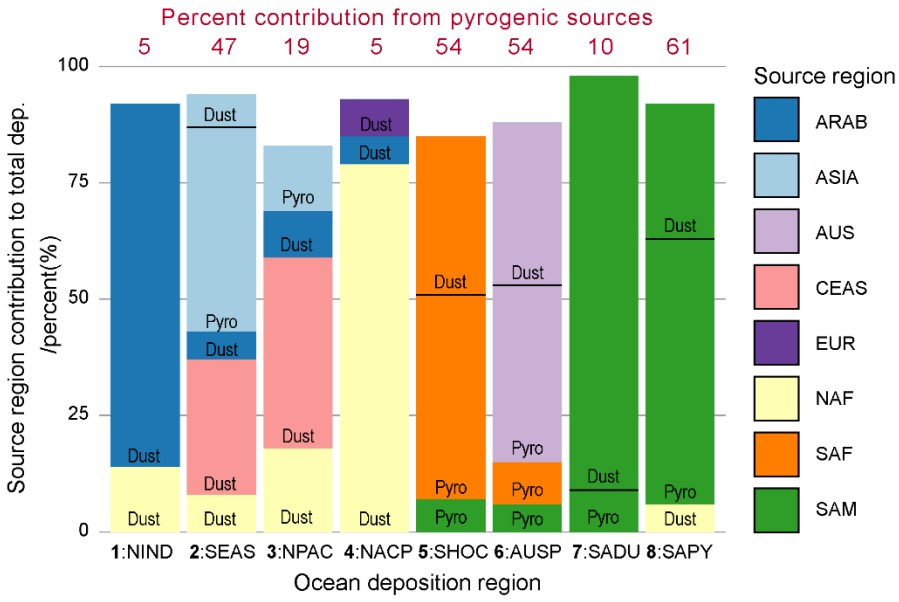


**Figure 14**. **Top.** Eight ocean soluble iron deposition regions defined by dominant source region apportionment. Region 1: North Indian Ocean (NIND). Region 2: South East Asian Ocean (SEAS). Region 3: North Pacific (NPAC). Region 4: North Atlantic and Central Pacific (NACP). Region 5: Southern Hemisphere Oceans (SHOC). 6: Australian and Southern Pacific (AUSP). 7: South America Dust (SADU). 8: South America Pyrogenic (SAPY). **Bottom.** Contribution of each emission source region (Fig. 1) to the total iron deposition across the region. Contribution of dust and pyrogenic (sum of fires and anthropogenic combustion) iron from source region also shown. Regions contributing <5% filtered out.

891

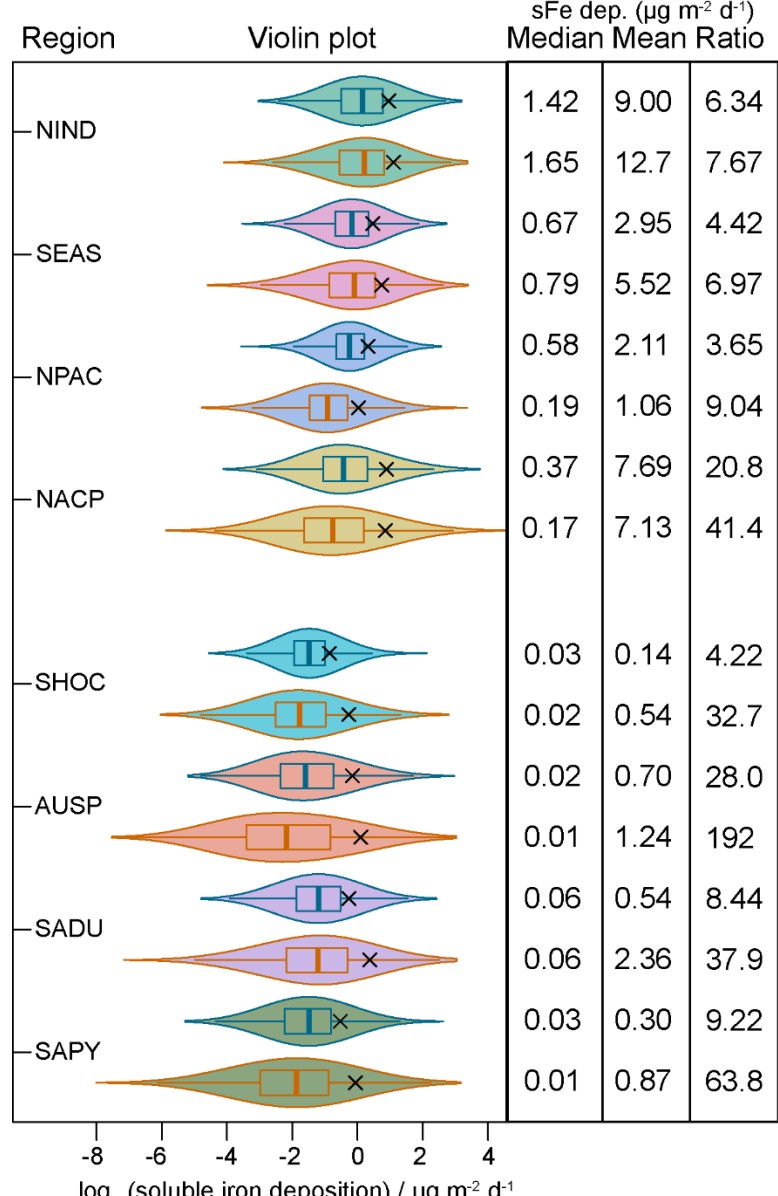

| Region | Violin plot | sFe dep. (µg m⁻² d⁻¹) Median | Mean | Ratio |
|---|---|---|---|---|
| NIND | | 1.42 | 9.00 | 6.34 |
| | | 1.65 | 12.7 | 7.67 |
| SEAS | | 0.67 | 2.95 | 4.42 |
| | | 0.79 | 5.52 | 6.97 |
| NPAC | | 0.58 | 2.11 | 3.65 |
| | | 0.19 | 1.06 | 9.04 |
| NACP | | 0.37 | 7.69 | 20.8 |
| | | 0.17 | 7.13 | 41.4 |
| SHOC | | 0.03 | 0.14 | 4.22 |
| | | 0.02 | 0.54 | 32.7 |
| AUSP | | 0.02 | 0.70 | 28.0 |
| | | 0.01 | 1.24 | 192 |
| SADU | | 0.06 | 0.54 | 8.44 |
| | | 0.06 | 2.36 | 37.9 |
| SAPY | | 0.03 | 0.30 | 9.22 |
| | | 0.01 | 0.87 | 63.8 |

$\log_{10}$(soluble iron deposition) / µg m⁻² d⁻¹

892

**Figure 15**. Violin plots of five years of $\log_{10}$ daily soluble iron deposition (µg m⁻² d⁻¹) within each grid cell for the eight ocean regions defined in Fig. 14. Only grid cells where ocean fraction >0.5 are included in analysis. Violin colour matches Fig. 1 region colour: North Indian Ocean (NIND); South East Asian Ocean (SEAS); North Pacific (NPAC); North Atlantic and Central Pacific (NACP); Southern Hemisphere Oceans (SHOC); Australian and Southern Pacific (AUSP); Southern American Dust (SADU); Southern American Combustion (SAPY). Violin outline colours: blue lines = BAM results while orange lines = MAM results. Black cross = $\log_{10}$ mean daily soluble iron deposition. Median, mean and ratio (mean/median) values for all five years of daily deposition amounts across each basin also given.

## 5 Future directions

The purpose of model to observation comparisons is to identify situations (regions, times, model settings, or combinations thereof) in which the model output is inconsistent with observed realities; with the goal being to further refine the model in the future. Each individual observation represents a snapshot of the atmospheric state at a specific point in space and time and when an observation falls outside of the distribution of model output values, from the same location and time, we can view this as evidence of a model misspecification. For the example of iron modelling, constraining current model-observation discrepancies would benefit from further exploring the model sensitivity of simulated iron and its solubility to uncertainties in five major parameter sets: dust iron emissions, pyrogenic iron emissions, atmospheric iron dissolution chemistry, dry deposition rates and wet deposition rates. In general, improving the modelled representation of secondary organic aerosol (including oxalate) and aerosol pH, particularly for remote regions, is an important task for aerosol modelling and one which would have co-benefits for iron aerosol modelling. Comparisons of soluble fraction of other aerosol species with observations could also be used to guide model development.

Here we discuss some of the model parameters which are likely important for improving modelled iron emissions and deposition in MIMI, and ergo iron process models in general, in the future.

## 5.1 Improving iron aerosol emissions

Downwind of significant mineral dust sources iron models generally overestimate the observed amount of total iron (Myriokefalitakis et al., 2018) and soluble iron comparisons are highly sensitive to the assumed initial solubility of mineral dust iron at emissions (Conway et al., 2019). Conversely in remote ocean regions, improving the representation of combustion emissions has been shown to be a necessary step towards more accurate representations of observed high iron solubilities at low iron concentrations (Ito et al., 2019).

### 5.1.1 Mineral dust iron aerosol emissions

In Fig. 4 the high model estimates of total iron, compared to observations, downwind of North African mineral dust sources could be due to uncertainties in the magnitude of hematite emissions within the model. Hematite contains by far the largest fraction of iron of any mineral in MIMI (Table 3) with a major source in the Sahel (Fig. S5). The Sahel is a borderline dust source and

emissions from this region have been shown to be sensitive to different model dynamics, even
when forced with reanalysis winds, for example between CAM4 and CAM5 (Scanza et al., 2015).
Other studies have shown a large sensitivity of dust generation to the details of the soil erodibility
map (e.g., Cakmur et al., 2006). For CAM5 with the DEAD emissions scheme Scanza et al. (2015)
showed that improvements in estimating the direct radiative forcing of mineral dust could be
achieved by assuming that hematite is only emitted from clay minerals and not silt, an effective
reduction of ~30% from the coarse mode emission of hematite. Although MIMI has employed an
updated dust emission scheme (Table 1; Kok et al., 2014a) the model is still sensitive to
assumptions within the offline minerology maps and applications of the brittle fragmentation
theory therein. For instance, the single scattering albedo, which is a critical parameter in
estimating the direct radiative forcing (e.g., Di Biagio et al., 2009), becomes more comparable to
observations (Kim et al., 2011) if the same assumption as Scanza et al. (2015) is applied (Fig.
S6). Quantifying the uncertainty on the climate response to different assumptions in minerology
and dust emissions, and any reanalysis meteorology driving them, is therefore an important task.

### 5.1.2 Pyrogenic iron aerosol emissions

Matsui et al. (2018) recently showed that combustion iron emissions have been underestimated
in current models. One possible reason for this underestimate is that anthropogenic combustion
iron emissions from Luo et al. (2008) are for 1996. Taking steelmaking and coal consumption
(which are also linked to iron emissions) as a proxy for economic development (Ghosh, 2006; Lee
and Chang, 2008) shows that growth in these sectors boomed exponentially post 2000,
particularly in Asia and India (Ghosh, 2006; Lee and Chang, 2008). Therefore 1996 emissions
are not capturing recent industrial developments and updating the anthropogenic combustion iron
emission inventory for use in the 21st Century is a critical next step.
During a fire, the iron contained in leaves and wood (Price, 1968) will be released to the
atmosphere along with iron contained in the surrounding soil, whether entrained from the ground
due to pyro-convective updrafts (Wagner et al., 2018) or a remobilization of terrigenous particles
which have previously been deposited onto vegetation (Gaudichet et al., 1995; Paris et al., 2010).
All sources are subsequently internally mixed within the smoke plume before any downwind
observation occurs. Differentiating the iron contribution from the biomass which is burnt to that
from the entrained dust was not considered in any of the studies in Table 4 but would be required
to define the correct minerology and solubility of iron from fires. If we assume that biomass
contains low concentrations of iron relative to the surrounding soils then we could expect a
difference in observed Fe:BC ratios between a cerrado (savannah) environment, where
surrounding soils are dry and dust is easily mobilized, compared to a tropical environment, where
soils are wet, and dust is not as easily mobilized. But we do not see this in Table 4, and both
regions have a similar range which spans around two orders of magnitude from low to high.
However, no concrete conclusions can be drawn from such a limited dataset and so more
observations are needed to distinguish which source (biomass or dust) is contributing most to the
iron measured downwind of fires.
The physical, chemical, and biological properties of the underlying soil are also impacted by fires
(Certini, 2005) and it can be years after the fire has occurred before returning to a pre-fire state
is achieved. For example, the removal of vegetation and the surface crust by fires from dune
regions will create a new opportunity for dust mobilization (Strong et al., 2010) and higher intensity
fires can also increase the erodibility of soils and availability of fine particles through breaking
down the soil structure (Levin et al., 2012). Furthermore, under high temperatures the fire can
transform the underlying soil minerology, with decreases to iron in clay minerals and increases in
magnetic iron oxide minerals (Crockford and Willett, 2001; Ketterings et al., 2000; Ulery and
Graham, 1993). The amount of dust emitted from post-fire landscapes is potentially very
significant with Wagenbrenner et al. (2017) estimating an extra 12-352 Tg of dust as $PM_{10}$ (40%
of which was estimated to be $PM_{2.5}$) was emitted to the atmosphere in 2012 from post-fire
landscapes in the western U.S. alone. The impact of fires on total and soluble iron emissions in
dust from within post-burn regions is also likely to be different but requires further study, although
likely depends on the fire regime and the time since the fire occurred.
The most advanced iron processing models currently consider industrial, domestic, wildfires and
shipping pyrogenic emissions (Myriokefalitakis et al., 2018). An emerging discussion is the
importance of volcanic ash, and the iron it contains, on ocean biogeochemistry (Langmann, 2013).
Figs. 4 through 7 showed that MIMI underrepresents both total iron and its solubility in the remote
extra-tropical Pacific where volcanic emissions may be an important missing iron source. Future
understanding in volcanic iron sources are potentially important as once deposited to the ocean,
particularly in those regions that are iron limited or seasonally iron limited, volcanic inputs have
been shown to alter satellite chlorophyll (Hamme et al., 2010; Rogan et al., 2016) and the
drawdown of macronutrients (Lindenthal et al., 2013). The volume of metals released by a volcano
is subject to many uncertainties, including both the nature of the volcano and its eruption type and
strength; leading to estimates which can vary by many orders of magnitude (Mather et al., 2006,
2012). To date most studies have focused on ocean inputs from shorter term explosive eruptions,
rather than continuous inputs from quiescent passive degassing volcanoes which are likely to be
most important only for the central Pacific region downwind of volcanoes located within the "ring
of fire" (Olgun et al., 2011).

## 5.2 Aerosol deposition

Examination of aerosol dry deposition in CAM5 by Wu et al (2018) showed that the deposition
velocity for Aitken and accumulation sized BC particles is potentially an order of magnitude too
high. It is highly likely that this will also be the case for dust. As the largest discrepancies between
model and observations are in remote ocean regions improving the models long-range transport
of iron by investigating deposition rates is an important constraint to be applied to the model.

## 6. Conclusion

It is important to accurately model the atmospheric iron cycle because of the impacts of iron on
human health, ocean biogeochemistry and climate. Atmospheric iron process modelling suitable
for use in global climate and Earth system modelling is a new model development area, and as
such currently undergoing rapid development. Here we have detailed the development of the
Mechanism of Intermediate complexity for Modelling Iron (MIMI v1.0), such that it now represents
iron emissions, atmospheric processing and deposition within a global modal aerosol
microphysics framework.
The solubility of iron depends on the underlying aerosol iron properties, such as dust mineralogy
and combustion fuel type, and the degree to which dissolution from an insoluble to soluble iron
form has occurred in the atmosphere. Which of these is the dominant factor for describing the
observed inverse relationship between the solubility of iron to the total iron mass is currently
unknown (Mahowald et al., 2018). Updating the mineral dust emission scheme to a physical
based parametrisation however has improved model performance by increasing total iron close
to mineral dust sources, where solubility is observed to be low (Figs. 4 through 7). Updating
pyrogenic iron emissions from fires increases the long range transport of soluble iron to remote
ocean regions, where observed solubility is higher (Figs. 4 through 7), while increasing
anthropogenic combustion iron emissions by a factor of five brings the total in line with more
recent evaluations of their magnitude (Conway et al., 2019; Matsui et al., 2018). Emission updates
have also increased the variability in soluble iron deposition (Fig. 15). Improvements to the
atmospheric iron processing scheme in MIMI also increase iron dissolution in more remote
regions relative to mineral dust sources, again in line with observations.
Comparison with observations (Figs. 4 through 7) show that in general MIMI simulates total iron
concentrations well. However, comparison of modelled iron solubility to observation reveals that
while the model captures many regional features, some are missed. It is unclear, however,
whether this problem arises from the model or observational representation of the system owing
to the insufficient numbers of observations available to build a robust observational result for such
a highly variable quantity in the Earth system, even when aggregating over small regional scales.
There are significant differences in calculating iron solubility based on the order of the averaging
operation. When calculating at each model time step global annual mean iron solubility is one-
third (34%; NH=40%, SH=29%) higher than when calculated from monthly mean values. Earth-
system models are designed to integrate land-atmosphere-ocean-ice components at each time-
step and thus could yield different results based on the coupling time-step length employed.
Furthermore, the mean is shown to not be an accurate representation of the average atmospheric
state, due to the non-Gaussian distribution of aerosol concentrations. In many regions however
there are just a few (less than five) observations, and often only one, and so while the use of the
median is robust with respect to extreme values, a limited observational dataset cannot truly
discriminate if extreme values are outliers or, rather, the norm. Use of the mean also significantly
overestimates the average atmospheric soluble iron deposition to the ocean and is always larger
than the upper quartile of the distribution in daily deposition. However, this bias may be tempered
due to ocean biogeochemistry processes likely being relevant over timescales which are longer
than those in the atmosphere. Future work will need to consider how best to compare model to
sporadic observations, potentially making use of distributions rather than a more limited absolute
average.
The main sources of soluble iron deposition vary both between and within ocean basis. The
redefinition of ocean basins based on the dominate iron deposition source, rather than a
traditional physio-geographical ocean basis, can therefore aid in determining where continental
anthropogenic activity will have the greatest impact on ocean biogeochemistry and which source
region is linked to where model-observation comparisons are poor. For example, modelling of
total iron and its solubility in the South Atlantic could be improved by further improving our
understanding of industrial combustion and fires within South America. Furthermore, soluble iron
deposition to Southern Hemisphere oceans in MIMI, where combustion and fire emissions have
a significant impact, is between a factor of two to four higher compared to BAM-Fe, itself the
model simulating the largest atmospheric fluxes to the ocean of the comparable models studied
in Myriokefalitakis et al. (2018). As integrated Earth system models develop in the future taking a
holistic view to understanding how dust and fires are coupled, in terms of feedbacks on iron
emissions, is an important step for predicting how future changes in climate will alter the climate
and Earth system response to human perturbations of the natural system.

**Code and data availability**

Model code (emissions and atmospheric processing for MIMI v1.0) and data is available at: http://www.geo.cornell.edu/eas/PeoplePlaces/Faculty/mahowald/dust/Hamiltonetal2019/

. Observational iron data is available from Mahowald et al. (2009) and Myriokefalitakis et al. (2018). Observational oxalate data is available from Myriokefalitakis et al. (2011).

**Author contributions**

D.S.H. developed MIMI which incorporates model code previously developed by R.A.S., Y.F., J.F.K., X.L., and M.W. D.S.H. undertook all model simulations and wrote the manuscript with support from N.M.M., J.G., and S.D.R.. D.S.H. prepared all Figures and Tables apart from Fig. 1 and Table S1 (J.S.W.), Figs. S3 and S6 (L.L.), and Fig. 9 and S2 (S.D.R.). All authors edited manuscript text.

**Acknowledgements**

This work was supported by Department of Energy (DE) and National Science Foundation (NSF) grants for atmospheric deposition impacts on ocean biogeochemistry (DE-Sc0016362; NSF 1049033; CCF-1522054). D.S.H. was also supported by the Atkinson Center for a Sustainable Future. J.F.K. acknowledges support from NSF grant 1552519. S.D.R. would like to thank the Collaborative Proposal Fire Dust Air and Water Improving Aerosol Biogeochemistry Interactions in ACME (DE-Sc0016321) in supporting his Masters. X.L. and M.W. would like to thank the support of NASA CloudSat and CALIPSO Science Program (grant NNX16AO94G). We would like to acknowledge high-performance computing support from Cheyenne (doi:10.5065/D6RX99HX) provided by NCAR's Computational and Information Systems Laboratory, sponsored by the National Science Foundation.

Achterberg, E. P., Moore, C. M., Henson, S. A., Steigenberger, S., Stohl, A., Eckhardt, S.,
Avendano, L. C., Cassidy, M., Hembury, D., Klar, J. K., Lucas, M. I., Macey, A. I., Marsay, C. M.
and Ryan-Keogh, Thomas, J.: Natural iron ferlilisation by the Eyjafjallajokull volcanic eruption,
Geophys. Res. Lett., 40, 921–926, doi:10.1002/grl.50221, 2013.
Akagi, S. K., Yokelson, R. J., Wiedinmyer, C., Alvarado, M. J., Reid, J. S., Karl, T., Crounse, J.
D. and Wennberg, P. O.: Emission factors for open and domestic biomass burning for use in
atmospheric models, Atmos. Chem. Phys., 11(9), 4039–4072, doi:10.5194/acp-11-4039-2011,

1106  2011.

Albani, S., Mahowald, N. M., Perry, A. T., Scanza, R. A., Zender, C. S., Heavens, N. G., Maggi,
V., Kok, J. F. and Otto-Bliesner, B. L.: Improved dust representation in the Community
Atmosphere Model, J. Adv. Model. Earth Sytems, 6(3), 541–570,
doi:10.1002/2013MS000279.Received, 2014.
Andreae, M. O. and Crutzen, P. J.: Atmospheric Aerosols: Biogeochemical Sources and Role in
Atmospheric Chemistry, Science, 276(5315), 1052–1058, doi:10.1126/science.276.5315.1052,

1113  1997.

Arimoto, R.: Eolian dust and climate: relationships to sources, tropospheric chemistry, transport
and deposition, Earth-Science Rev., 54(1–3), 29–42, doi:10.1016/S0012-8252(01)00040-X,

1116  2001.

Artaxo, P., Rizzo, L. V., Brito, J. F., Barbosa, H. M. J., Arana, A., Sena, E. T., Cirino, G. G.,
Bastos, W., Martin, S. T. and Andreae, M. O.: Atmospheric aerosols in Amazonia and land use
change: from natural biogenic to biomass burning conditions, Faraday Discuss.,
doi:10.1039/c3fd00052d, 2013.
Aumont, O., Ethé, C., Tagliabue, A., Bopp, L. and Gehlen, M.: PISCES-v2: An ocean
biogeochemical model for carbon and ecosystem studies, Geosci. Model Dev., 8(8), 2465–
2513, doi:10.5194/gmd-8-2465-2015, 2015.
Baker, A. R., Jickells, T. D., Biswas, K. F., Weston, K. and French, M.: Nutrients in atmospheric
aerosol particles along the Atlantic Meridional Transect, Deep Sea Res. Part II Top. Stud.
Oceanogr., 53(14–16), 1706–1719, doi:10.1016/j.dsr2.2006.05.012, 2006a.
Baker, A. R., Jickells, T. D., Witt, M. and Linge, K. L.: Trends in the solubility of iron, aluminium,
manganese and phosphorus in aerosol collected over the Atlantic Ocean, Mar. Chem., 98(1),
43–58, doi:10.1016/j.marchem.2005.06.004, 2006b.
Bates, T. S., Lamb, B. K., Guenther, A., Dignon, J. and Stoiber, R. E.: Sulfur Emissions to the
Atmosphere from Natural Sources, J. Atmos. Chem., 14, 315–337, 1992.
Di Biagio, C., Di Sarra, A., Meloni, D., Monteleone, F., Piacentino, S. and Sferlazzo, D.:
Measurements of Mediterranean aerosol radiative forcing and influence of the single scattering
albedo, J. Geophys. Res. Atmos., 114(6), 1–12, doi:10.1029/2008JD011037, 2009.
Böke, H., Göktürk, E. H., Caner-Saltık, E. N. and Demirci, Ş.: Effect of airborne particle on
SO2–calcite reaction, Appl. Surf. Sci., 140(1–2), 70–82, doi:10.1016/S0169-4332(98)00468-1,

1137    1999.

Boyd, P. W., Jickells, T., Law, C. S., Blain, S., Boyle, E. A., Buesseler, K. O., Coale, K. H.,
Cullen, J. J., Baar, H. J. W. De, Follows, M., Harvey, M., Lancelot, C., Levasseur, M., Owens, N.
P. J., Pollard, R., Rivkin, R. B., Sarmiento, J., Schoemann, V., Smetacek, V., Takeda, S.,
Tsuda, A., Turner, S. and Watson, A. J.: Mesoscale Iron Enrichment Experiments 1993 – 2005:
Synthesis and Future Directions, Science, 315, 612–618, 2007.
Bullard, J. E.: The distribution and biogeochemical importance of highlatitude dust in the Arctic
and Southern Ocean- Antarctic regions, J. Geophys. Res., 122(5), 3098–3103,
doi:10.1002/2016JD026363, 2017.
Bullard, J. E., Baddock, M., Bradwell, T., Crusius, J., Darlington, E., Gaiero, D., Gassó, S.,
Gisladottir, G., Hodgkins, R., McCulloch, R., McKenna-Neuman, C., Mockford, T., Stewart, H.
and Thorsteinsson, T.: High-latitude dust in the Earth system, Rev. Geophys., 54(2), 447–485,
doi:10.1002/2016RG000518, 2016.
Cakmur, R. V., Miller, R. L., Perlwitz, J., Geogdzhayev, I. V., Ginoux, P., Koch, D., Kohfeld, K.
E., Tegen, I. and Zender, C. S.: Constraining the magnitude of the global dust cycle by
minimizing the difference between a model and observations, J. Geophys. Res. Atmos., 111(6),
1–24, doi:10.1029/2005JD005791, 2006.
Capone, D., Zehr, J., Paerl, H., Bergman, B. and Carpenter, E.: Trichodesmium, a globallly
significant marine cyanobacterium, Science (80-. )., 276, 1221–1229, 1997.
Certini, G.: Effects of fire on properties of forest soils: A review, Oecologia, 143(1), 1–10,
doi:10.1007/s00442-004-1788-8, 2005.
Chin, M. and Jacob, D. J.: Anthropogenic and natural contributions to tropospheric sulfate: A
global model analysis, J. Geophys. Res. Atmos., 101(D13), 18691–18699,
doi:10.1029/96JD01222, 1996.
Chuang, P. Y., Duvall, R. M., Shafer, M. M. and Schauer, J. J.: The origin of water soluble
particulate iron in the Asian atmospheric outflow, Geophys. Res. Lett., 32(7), 1–4,
doi:10.1029/2004GL021946, 2005.
Claquin, T., Schulz, M. and Balkanski, Y. J.: Modeling the Minerology of Atmospheric Dust
Sources, J. Geophys. Res. Res., 104(D18), 22243–22256, 1999.
Computational and Information Systems Laboratory: Cheyenne: HPE/SGI ICE XA System
(University Community Computing), Boulder, CO Natl. Cent. Atmos. Res.,
doi:10.5065/D6RX99HX, 2017.
Conway, T. M., Hamilton, D. S., Shelley, R. U., Aguilar-Islas, A. M., Landing, W. M., Mahowald,
N. M. and John, S. G.: Tracing and constraining anthropogenic aerosol iron fluxes to the North
Atlantic Ocean using iron isotopes, Nat. Commun., 10(1), 1–10, doi:10.1038/s41467-019-
10457-w, 2019.
Cornell, R. and Schindler, P.: Photochemical dissolution of goethite in acid/oxalate solution,
Clays Clay Miner., 35(5), 347–352, doi:10.1346/CCMN.1987.0350504, 1987.
Crockford, R. H. and Willett, I. R.: Application of mineral magnetism to describe profile
development of toposequences of a sedimentary soil in south-eastern Australia, Aust. J. Soil
Res., 39(5), 927–949, doi:10.1071/SR00077, 2001.
Crusius, J., Schroth, A. W., Gassó, S., Moy, C. M., Levy, R. C. and Gatica, M.: Glacial flour dust
storms in the Gulf of Alaska: Hydrologic and meteorological controls and their importance as a
source of bioavailable iron, Geophys. Res. Lett., 38(L06602), 1–5, doi:10.1029/2010GL046573,

1181    2011.

Dentener, F., Kinne, S., Bond, T., Boucher, O., Cofala, J., Generoso, S., Ginoux, P., Gong, S.,
Hoelzemann, J. J., Ito, A., Marelli, L., Penner, J. E., Putaud, J.-P., Textor, C., Schultz, M., van
der Werf, G. R. and Wilson, J.: Emissions of primary aerosol and precursor gases in the years
2000 and 1750 prescribed data-sets for AeroCom, Atmos. Chem. Phys., 6, 4321–4344, 2006.
Duce, R. and Tindale, N.: Atmospheric transport of iron and its deposition in the ocean, Limnol.
Ocean., 36(8), 1715–1726, 1991.
Facchini, M. C., Rinaldi, M., Decesari, S., Carbone, C., Finessi, E., Mircea, M., Fuzzi, S.,
Ceburnis, D., Flanagan, R., Nilsson, E. D., de Leeuw, G., Martino, M., Woeltjen, J. and O'Dowd,

C. D.: Primary submicron marine aerosol dominated by insoluble organic colloids and

aggregates, Geophys. Res. Lett., 35(17), 1–5, doi:10.1029/2008GL034210, 2008.

Fanourgakis, G. S., Kanakidou, M., Nenes, A., Bauer, S. E., Bergman, T., Carslaw, K. S., Grini,

A., Hamilton, D. S., Johnson, J. S., Karydis, V. A., Kirkevåg, A., Kodros, J. K., Lohmann, U.,

Luo, G., Makkonen, R., Matsui, H., Neubauer, D., Pierce, J. R., Schmale, J., Stier, P.,

Tsigaridis, K., van Noije, T., Wang, H., Watson-Parris, D., Westervelt, D. M., Yang, Y.,

Yoshioka, M., Daskalakis, N., Decesari, S., Gysel-Beer, M., Kalivitis, N., Liu, X., Mahowald, N.

M., Myriokefalitakis, S., Schrödner, R., Sfakianaki, M., Tsimpidi, A. P., Wu, M. and Yu, F.:

Evaluation of global simulations of aerosol particle and cloud condensation nuclei number, with

implications for cloud droplet formation, Atmos. Chem. Phys., 19(13), 8591–8617,

doi:10.5194/acp-19-8591-2019, 2019.

Fung, I., Meyn, S. K., Tegen, I., Doney, S., John, J. and Bishop, J.: Iron supply and demand in

the upper ocean, Global Biogeochem. Cycles, 14(1), 281–295, 2000.

Gaudichet, A., Echalar, F., Chatenet, B., Quisefit, J. P., Malingre, G., Cachier, H., Buat-Menard,

P., Artaxo, P. and Maenhaut, W.: Trace elements in tropical African savanna biomass burning

aerosols, J. Atmos. Chem., 22(1–2), 19–39, doi:10.1007/BF00708179, 1995.

Gettelman, A., Liu, X., Ghan, S. J., Morrison, H., Park, S., Conley, A. J., Klein, S. A., Boyle, J.,

Mitchell, D. L. and Li, J. L. F.: Global simulations of ice nucleation and ice supersaturation with

an improved cloud scheme in the Community Atmosphere Model, J. Geophys. Res. Atmos.,

115(18), 1–19, doi:10.1029/2009JD013797, 2010.

Ghosh, S.: Steel consumption and economic growth: Evidence from India, Resour. Policy,

31(1), 7–11, doi:10.1016/j.resourpol.2006.03.005, 2006.

Giglio, L., Randerson, J. T. and van der Werf, G. R.: Analysis of daily, monthly, and annual

burned area using the fourth-generation global fire emissions database (GFED4), J. Geophys.

Res. Biogeosciences, 118(1), 317–328, doi:10.1002/jgrg.20042, 2013.

Golaz, J., Caldwell, P. M., Roekel, L. P. Van, Petersen, M. R., Tang, Q., Wolfe, J. D., Abeshu,

G., Anantharaj, V., Asay-davis, X. S., Bader, D. C., Baldwin, S. A., Bisht, G., Bogenschutz, P.

A., Branstetter, M., Brunke, M. A., Brus, S. R., Burrows, S. M., Cameron-smith, P. J., Donahue,

A. S., Deakin, M., Easter, R. C., Evans, K. J., Feng, Y., Flanner, M., Foucar, J. G., Fyke, J. G.,

Hunke, E. C., Jacob, R. L., Jacobsen, D. W., Jeffery, N., Jones, P. W., Keen, N. D., Klein, S. A.,

Larson, V. E., Leung, L. R., Li, H., Lin, W., Lipscomb, W. H., Ma, P., Mccoy, R. B., Neale, R. B.,

Price, S. F., Qian, Y., Rasch, P. J., Eyre, J. E. J. R., Riley, W. J., Ringler, T. D., Roberts, A. F.,
Roesler, E. L., Salinger, A. G., Shaheen, Z., Shi, X., Singh, B., Veneziani, M., Wan, H., Wang,
H., Wang, S., Williams, D. N., Wolfram, P. J., Worley, P. H., Xie, S., Yang, Y., Yoon, J.-H.,
Zelinka, M. D., Zender, C. S., Zeng, X., Zhang, C., Zhang, K., Zhang, Y., Zheng, X., Zhou, T.
and Zhu, Q.: The DOE E3SM Coupled Model Version 1: Overview and Evaluation at Standard
Resolution, J. Adv. Model. Earth Sytems, 11, 1–41, doi:10.1029/2018MS001603, 2019.
Gregg, W. W., Conkright, M. E., Ginoux, P., O'Reilly, J. E. and Casey, N. W.: Ocean primary
production and climate: Global decadal changes, Geophys. Res. Lett., 30(15), 10–13,
doi:10.1029/2003GL016889, 2003.
Guieu, C., Bonnet, S., Wagener, T. and Loÿe-Pilot, M. D.: Biomass burning as a source of
dissolved iron to the open ocean?, Geophys. Res. Lett., 32(L19608), 1–5,
doi:10.1029/2005GL022962, 2005.
Guieu, C., Aumont, O., Paytan, A., Bopp, L., Law, C. S., Mahowald, N., Achterberg, E. P.,
Marañón, E., Salihoglu, B., Crise, A., Wagener, T., Herut, B., Desboeufs, K., Kanakidou, M.,
Olgun, N., Peters, F., Völker, C., Aumont, O., Paytan, A., Bopp, L., Law, C. S., Mahowald, N.,
Achterberg, E. P., Marañón, E., Salihoglu, B., Crise, A., Wagener, T., Herut, B., Desboeufs, K.,
Kanakidou, M., Olgun, N., Peters, F. and Völker, C.: Global Biogeochemical Cycles deposition
to Low Nutrient Low Chlorophyll regions, Global Biogeochem. Cycles, 28, 1179–1198,
doi:10.1002/2014GB004852.Received, 2014.
Hamme, R., Webley, P., Crawford, W., Whitney, F., DeGrandpre, M., Emerson, S., Eriksen, C.,
Giesbrecht, K., Gower, J., Kavanaugh, M., Peña, M., Sabine, C., Batten, S., Coogan, L.,
Grundle, D. and Lockwood, D.: Volcanic ash fuels anomalous plankton bloom in subarctic
northeast Pacific, Geophys. Res. Lett., 37(L19604), 1–5, doi:10.1029/2010GL044629, 2010.
Holben, B. N., Eck, T. F., Slutsker, I., Tanre, D., Cimel, J. P. B., Vermote, E., Reagan, J. A.,
Kaufman, Y. J., Nakajima, T., Lavenu, F., Jankowiak, I. and Smirnov, A.: AERONET-A
Federated Instrument Network and Data Archeive for Aerosol Characterization, 2000.
Hu, M., Peng, J., Sun, K., Yue, D., Guo, S., Wiedensohler, A. and Wu, Z.: Estimation of size-
resolved ambient particle density based on the measurement of aerosol number, mass, and
chemical size distributions in the winter in Beijing, Environ. Sci. Technol., 46(18), 9941–9947,
doi:10.1021/es204073t, 2012.
Huneeus, N., Schulz, M., Balkanski, Y., Griesfeller, J., Prospero, J., Kinne, S., Bauer, S.,
Boucher, O., Chin, M., Dentener, F., Diehl, T., Easter, R., Fillmore, D., Ghan, S., Ginoux, P.,
Grini, A., Horowitz, L., Koch, D., Krol, M. C., Landing, W., Liu, X., Mahowald, N., Miller, R.,
Morcrette, J.-J., Myhre, G., Penner, J., Perlwitz, J., Stier, P., Takemura, T. and Zender, C. S.:
Global dust model intercomparison in AeroCom phase I, Atmos. Chem. Phys., 11(15), 7781–
7816, doi:10.5194/acp-11-7781-2011, 2011.
Ingall, E., Feng, Y., Longo, A., Lai, B., Landing, W., Shelley, R., Morton, P., Nenes, A., Violaki,
K., Gao, Y., Sahai, S. and Castorina, E.: Enhanced Iron Solubility at Low pH in Global Aerosols,
Atmosphere (Basel)., 9(5), 201, doi:10.3390/atmos9050201, 2018.
Ito, A.: Global modeling study of potentially bioavailable iron input from shipboard aerosol
sources to the ocean, Global Biogeochem. Cycles, 27(1), 1–10, doi:10.1029/2012GB004378,
1262 2013.

Ito, A.: Atmospheric processing of combustion aerosols as a source of bioavailable iron,
Environ. Sci. Technol. Lett., 2(3), 70–75, doi:10.1021/acs.estlett.5b00007, 2015.
Ito, A. and Xu, L.: Response of acid mobilization of iron-containing mineral dust to improvement
of air quality projected in the future, Atmos. Chem. Phys., 14(7), 3441–3459, doi:10.5194/acp-
1267 14-3441-2014, 2014.

Ito, A., Myriokefalitakis, S., Kanakidou, M., Mahowald, N. M., Scanza, R. A., Hamilton, D. S.,
Baker, A. R., Jickells, T., Sarin, M., Bikkina, S., Gao, Y., Shelley, R. U., Buck, C. S., Landing, W.
M., Bowie, A. R., Perron, M. M. G., Guieu, C., Meskhidze, N., Johnson, M. S., Feng, Y., Kok, J.
F., Nenes, A. and Duce, R. A.: Pyrogenic iron: The missing link to high iron solubility in
aerosols, Sci. Adv., 5(5), 1–10, doi:10.1126/sciadv.aau7671, 2019.
Jickells, T., Boyd, P. and Hunter, K.: Biogeochemical impacts of dust on the global carbon cycle,
in Mineral Dust: A Key player in the Earth System, edited by P. Knippertz and J.-B. Stutt, pp.
284–359, Springer Science+ Business Media, Dordrecht., 2014.
Jickells, T. D., An, Z. S., Andersen, K. K., Baker,  a R., Bergametti, G., Brooks, N., Cao, J. J.,
Boyd, P. W., Duce, R. a, Hunter, K. a, Kawahata, H., Kubilay, N., LaRoche, J., Liss, P. S.,
Mahowald, N., Prospero, J. M., Ridgwell,  a J., Tegen, I. and Torres, R.: Global iron connections
between desert dust, ocean biogeochemistry, and climate., Science, 308(5718), 67–71,
doi:10.1126/science.1105959, 2005.
Johnson, M. S. and Meskhidze, N.: Atmospheric dissolved iron deposition to the global oceans:
Effects of oxalate-promoted Fe dissolution, photochemical redox cycling, and dust mineralogy,
Geosci. Model Dev., 6(4), 1137–1155, doi:10.5194/gmd-6-1137-2013, 2013.
Journet, E., Desbouefs, K., Caqineau, S. and Colin, J.-L.: Mineralogy as a critical factor of dust
iron solubility, Geophys. Res. Lett., 35(L07805), doi:10.1029/2007GL031589, 2008.
Kanakidou, M., Seinfeld, J. H., Pandis, S. N., Barnes, I., Dentener, F. J., Facchini, M. C., Van
Dingenen, R., Ervens, B., Nenes, A., Nielsen, C. J., Swietlicki, E., Putaud, J. P., Balkanski, Y.,
Fuzzi, S., Horth, J., Moortgat, G. K., Winterhalter, R., Myhre, C. E. L., Tsigaridis, K., Vignati, E.,
Stephanou, E. G. and Wilson, J.: Organic aerosol and global climate modelling: a review,
Atmos. Chem. Phys., 5(4), 1053–1123, doi:10.5194/acp-5-1053-2005, 2005.
Ketterings, Q. M., Bigham, J. M. and Laperche, V.: Changes in Soil Mineralogy and Texture
Caused by Slash-and-Burn Fires in Sumatra, Indonesia, Soil Sci. Soc. Am. J., 64(3), 1108–
1117, doi:10.2136/sssaj2000.6431108x, 2000.
Kim, D., Chin, M., Yu, H., Eck, T. F., Sinyuk, A., Smirnov, A. and Holben, B. N.: Dust optical
properties over North Africa and Arabian Peninsula derived from the AERONET dataset, Atmos.
Chem. Phys., 11(20), 10733–10741, doi:10.5194/acp-11-10733-2011, 2011.
Kok, J. F.: A scaling theory for the size distribution of emitted dust aerosols suggests climate
models underestimate the size of the global dust cycle, Proc. Natl. Acad. Sci., 108(3), 1016–
1021, doi:10.1073/pnas.1014798108, 2011.
Kok, J. F., Mahowald, N. M., Fratini, G., Gillies, J. A., Ishizuka, M., Leys, J. F., Mikami, M., Park,
M. S., Park, S. U., Van Pelt, R. S. and Zobeck, T. M.: An improved dust emission model - Part
1: Model description and comparison against measurements, Atmos. Chem. Phys., 14(23),
13023–13041, doi:10.5194/acp-14-13023-2014, 2014a.
Kok, J. F., Albani, S., Mahowald, N. M. and Ward, D. S.: An improved dust emission model -
Part 2: Evaluation in the Community Earth System Model, with implications for the use of dust
source functions, Atmos. Chem. Phys., 14(23), 13043–13061, doi:10.5194/acp-14-13043-2014,
2014b.
Kok, J. F., Ridley, D. A., Zhou, Q., Miller, R. L., Zhao, C., Heald, C. L., Ward, D. S., Albani, S.
and Haustein, K.: Smaller desert dust cooling effect estimated from analysis of dust size and
abundance, Nat. Geosci., 10(4), 274–278, doi:10.1038/ngeo2912, 2017.
Lamarque, J.-F., Bond, T. C., Eyring, V., Granier, C., Heil, A., Klimont, Z., Lee, D., Liousse, C.,
Mieville, A., Owen, B., Schultz, M. G., Shindell, D., Smith, S. J., Stehfest, E., Van Aardenne, J.,
Cooper, O. R., Kainuma, M., Mahowald, N., McConnell, J. R., Naik, V., Riahi, K. and van
Vuuren, D. P.: Historical (1850–2000) gridded anthropogenic and biomass burning emissions of
reactive gases and aerosols: methodology and application, Atmos. Chem. Phys., 10(15), 7017–
7039, doi:10.5194/acp-10-7017-2010, 2010.
Langmann, B.: Volcanic Ash versus Mineral Dust: Atmospheric Processing and Environmental
and Climate Impacts, ISRN Atmos. Sci., Article ID, 1–17, doi:10.1155/2013/245076, 2013.
Langmann, B., Zakšek, K., Hort, M. and Duggen, S.: Volcanic ash as fertiliser for the surface
ocean, Atmos. Chem. Phys., 10, 3891–3899, 2010.
Lee, C. C. and Chang, C. P.: Energy consumption and economic growth in Asian economies: A
more comprehensive analysis using panel data, Resour. Energy Econ., 30(1), 50–65,
doi:10.1016/j.reseneeco.2007.03.003, 2008.
Levin, N., Levental, S. and Morag, H.: The effect of wildfires on vegetation cover and dune
activity in Australia's desert dunes: A multisensor analysis, Int. J. Wildl. Fire, 21(4), 459–475,
doi:10.1071/WF10150, 2012.
Li, F., Koopal, L. and Tan, W.: Roles of different types of oxalate surface complexes in
dissolution process of ferrihydrite aggregates, Sci. Rep., 8(1), 1–13, doi:10.1038/s41598-018-

1329 20401-5, 2018.

Lindenthal, A., Langmann, B., Pätsch, J., Lorkowski, I. and Hort, M.: The ocean response to
volcanic iron fertilisation after the eruption of Kasatochi volcano: A regional-scale
biogeochemical ocean model study, Biogeosciences, 10(6), 3715–3729, doi:10.5194/bg-10-

1333 3715-2013, 2013.

Liu, X., Easter, R. C., Ghan, S. J., Zaveri, R., Rasch, P., Shi, X., Lamarque, J. F., Gettelman, A.,
Morrison, H., Vitt, F., Conley, A., Park, S., Neale, R., Hannay, C., Ekman, A. M. L., Hess, P.,
Mahowald, N., Collins, W., Iacono, M. J., Bretherton, C. S., Flanner, M. G. and Mitchell, D.:
Toward a minimal representation of aerosols in climate models: Description and evaluation in
the Community Atmosphere Model CAM5, Geosci. Model Dev., 5(3), 709–739,
doi:10.5194/gmd-5-709-2012, 2012.
Liu, X., Ma, P. L., Wang, H., Tilmes, S., Singh, B., Easter, R. C., Ghan, S. J. and Rasch, P. J.:
Description and evaluation of a new four-mode version of the Modal Aerosol Module (MAM4)
within version 5.3 of the Community Atmosphere Model, Geosci. Model Dev., 9(2), 505–522,
doi:10.5194/gmd-9-505-2016, 2016.

Longo, A. F., Feng, Y., Lai, B., Landing, W. M., Shelley, R. U., Nenes, A., Mihalopoulos, N., Violaki, K. and Ingall, E. D.: Influence of Atmospheric Processes on the Solubility and Composition of Iron in Saharan Dust, Environ. Sci. Technol., 50(13), 6912–6920, doi:10.1021/acs.est.6b02605, 2016.

Luo, C., Mahowald, N., Bond, T., Chuang, P. Y., Artaxo, P., Siefert, R., Chen, Y. and Schauer, J.: Combustion iron distribution and deposition, Global Biogeochem. Cycles, 22(GB1012), 1–17, doi:10.1029/2007GB002964, 2008.

Mahowald, N.: Aerosol indirect effect on biogeochemical cycles and climate., Science, 334(6057), 794–6, doi:10.1126/science.1207374, 2011.

Mahowald, N., Jickells, T. D., Baker, A. R., Artaxo, P., Benitez-Nelson, C. R., Bergametti, G., Bond, T. C., Chen, Y., Cohen, D. D., Herut, B., Kubilay, N., Losno, R., Luo, C., Maenhaut, W., McGee, K. A., Okin, G. S., Siefert, R. L. and Tsukuda, S.: Global distribution of atmospheric phosphorus sources, concentrations and deposition rates, and anthropogenic impacts, Global Biogeochem. Cycles, 22(4), 1–19, doi:10.1029/2008GB003240, 2008.

Mahowald, N. M., Engelstaedter, S., Luo, C., Sealy, A., Artaxo, P., Benitez-Nelson, C., Bonnet, S., Chen, Y., Chuang, P. Y., Cohen, D. D., Dulac, F., Herut, B., Johansen, A. M., Kubilay, N., Losno, R., Maenhaut, W., Paytan, A., Prospero, J. M., Shank, L. M. and Siefert, R. L.: Atmospheric iron deposition: global distribution, variability, and human perturbations., Ann. Rev. Mar. Sci., 245–278, doi:10.1146/annurev.marine.010908.163727, 2009.

Mahowald, N. M., Scanza, R., Brahney, J., Goodale, C. L., Hess, P. G., Moore, J. K. and Neff, J.: Aerosol Deposition Impacts on Land and Ocean Carbon Cycles, Curr. Clim. Chang. Reports, 3(1), 16–31, doi:10.1007/s40641-017-0056-z, 2017.

Mahowald, N. M., Hamilton, D. S., Mackey, K. R. M., Moore, J. K., Baker, A. R., Scanza, R. A. and Zhang, Y.: Aerosol trace metal leaching and impacts on marine microorganisms, Nat. Commun., 9(1), doi:10.1038/s41467-018-04970-7, 2018.

Mann, G. W., Carslaw, K. S., Reddington, C. L., Pringle, K. J., Schulz, M., Asmi, A., Spracklen, D. V., Ridley, D. a., Woodhouse, M. T., Lee, L. a., Zhang, K., Ghan, S. J., Easter, R. C., Liu, X., Stier, P., Lee, Y. H., Adams, P. J., Tost, H., Lelieveld, J., Bauer, S. E., Tsigaridis, K., van Noije, T. P. C., Strunk, A., Vignati, E., Bellouin, N., Dalvi, M., Johnson, C. E., Bergman, T., Kokkola, H., von Salzen, K., Yu, F., Luo, G., Petzold, A., Heintzenberg, J., Clarke, A., Ogren, J. a., Gras, J., Baltensperger, U., Kaminski, U., Jennings, S. G., O'Dowd, C. D., Harrison, R. M., Beddows,

D. C. S., Kulmala, M., Viisanen, Y., Ulevicius, V., Mihalopoulos, N., Zdimal, V., Fiebig, M., Hansson, H.-C., Swietlicki, E. and Henzing, J. S.: Intercomparison and evaluation of global aerosol microphysical properties among AeroCom models of a range of complexity, Atmos. Chem. Phys., 14(9), 4679–4713, doi:10.5194/acp-14-4679-2014, 2014.

Martin, H., Gordon, R. M. and Fitzwater, S. E.: The case for iron, Limnol. Ocean., 36(8), 1793–1802, 1991.

Martin, J.: Glacial-interglacial CO2 change: The iron hypothesis, Paleoceanography, 5(1), 1–13, 1990.

Mather, T. A., Pyle, D. M., Tsanev, V. I., McGonigle, A. J. S., Oppenheimer, C. and Allen, A. G.: A reassessment of current volcanic emissions from the Central American arc with specific examples from Nicaragua, J. Volcanol. Geotherm. Res., 149(3–4), 297–311, doi:10.1016/j.jvolgeores.2005.07.021, 2006.

Mather, T. A., Witt, M. L. I., Pyle, D. M., Quayle, B. M., Aiuppa, A., Bagnato, E., Martin, R. S., Sims, K. W. W., Edmonds, M., Sutton, A. J. and Ilyinskaya, E.: Halogens and trace metal emissions from the ongoing 2008 summit eruption of Kīlauea volcano, Hawaìi, Geochim. Cosmochim. Acta, 83, 292–323, doi:10.1016/j.gca.2011.11.029, 2012.

Matsui, H., Mahowald, N. M., Moteki, N., Hamilton, D. S., Ohata, S., Yoshida, A., Koike, M., Scanza, R. A. and Flanner, M. G.: Anthropogenic combustion iron as a complex climate forcer, Nat. Commun., 9(1), doi:10.1038/s41467-018-03997-0, 2018.

Menut, L., Bessagnet, B., Colette, A. and Khvorostiyanov, D.: On the impact of the vertical resolution on chemistry-transport modelling, Atmos. Environ., 67, 370–384, doi:10.1016/j.atmosenv.2012.11.026, 2013.

Meskhidze, N., Chameides, W. L., Nenes, A. and Chen, G.: Iron mobilization in mineral dust: Can anthropogenic SO2 emissions affect ocean productivity?, Geophys. Res. Lett., 30(21), 2085, doi:10.1029/2003GL018035, 2003.

Meskhidze, N., Chameides, W. L. and Nenes, A.: Dust and pollution: A recipe for enhanced ocean fertilization?, J. Geophys. Res. Atmos., 110(D03301), 1–23, doi:10.1029/2004JD005082, 2005.

Moore, C. M. M., Mills, M. M. M., Arrigo, K. R. R., Berman-Frank, I., Bopp, L., Boyd, P. W. W., Galbraith, E. D. D., Geider, R. J. J., Guieu, C., Jaccard, S. L. L., Jickells, T. D. D., La Roche, J.,

Lenton, T. M. M., Mahowald, N. M. M., Marañón, E., Marinov, I., Moore, J. K. K., Nakatsuka, T.,
Oschlies, A., Saito, M. A. A., Thingstad, T. F. F., Tsuda, A. and Ulloa, O.: Processes and
patterns of oceanic nutrient limitation, Nat. Geosci., 6(9), 701–710, doi:10.1038/ngeo1765,
1408 2013.

Moore, J. K., Doney, S. C. and Lindsay, K.: Upper ocean ecosystem dynamics and iron cycling
in a global three-dimensional model, Global Biogeochem. Cycles, 18(4), 1–21,
doi:10.1029/2004GB002220, 2004.
Moore, K., Doney, S. C., Lindsay, K., Mahowald, N. and Michaels Anthony F., A. F.: Nitrogen
fixation amplifies the ocean biogeochemical response to decadal timescale variations in mineral
dust deposition, Tellus, Ser. B Chem. Phys. Meteorol., 58(5), doi:10.1111/j.1600-
0889.2006.00209.x, 2006.
Morrison, H. and Gettelman, A.: A new two-moment bulk stratiform cloud microphysics scheme
in the community atmosphere model, version 3 (CAM3). Part I: Description and numerical tests,
J. Clim., 21(15), 3642–3659, doi:10.1175/2008JCLI2105.1, 2008.
Myriokefalitakis, S., Tsigaridis, K., Mihalopoulos, N., Sciare, J., Nenes, A., Kawamura, K.,
Segers, A. and Kanakidou, M.: In-cloud oxalate formation in the global troposphere: A 3-D
modeling study, Atmos. Chem. Phys., 11(12), 5761–5782, doi:10.5194/acp-11-5761-2011,
1422 2011.

Myriokefalitakis, S., Daskalakis, N., Mihalopoulos, N., Baker, A. R., Nenes, A. and Kanakidou,
M.: Changes in dissolved iron deposition to the oceans driven by human activity: a 3-D global
modelling study, Biogeosciences, 12(13), 3973–3992, doi:10.5194/bg-12-3973-2015, 2015.
Myriokefalitakis, S., Ito, A., Kanakidou, M., Nenes, A., Krol, M. C., Mahowald, N. M., Scanza, R.
A., Hamilton, D. S., Johnson, M. S., Meskhidze, N., Kok, J. F., Guieu, C., Baker, A. R., Jickells,
T. D., Sarin, M. M., Bikkina, S., Shelley, R., Bowie, A., Perron, M. M. G. and Duce, R. A.:
Reviews and syntheses: The GESAMP atmospheric iron deposition model intercomparison
study, Biogeosciences, 15(21), 6659–6684, doi:10.5194/bg-15-6659-2018, 2018.
Neale, R. B., Chen, C. C., Gettelman, A., Lauritzen, P. H., Park, S., Williamson, D. L., Conley,
A. J., Garcia, R., Kinnison, D., Lamarque, J. F., Marsh, D., Mills, M., Smith, A. K., Tilmes, S.,
Vitt, F., Morrison, H., Cameron-Smith, P., Collins, W. D., Iacono, M. J., Easter, R. C., Ghan, S.
J., Liu, X., Rasch, P. J. and Taylor, M. A.: Description of the NCAR Community Atmosphere
Model (CAM 5.0)., 2010.

O'Dowd, C. D. and de Leeuw, G.: Marine aerosol production: a review of the current knowledge, Philos. Trans. R. Soc. A Math. Phys. Eng. Sci., 365(1856), 1753–1774, doi:10.1098/rsta.2007.2043, 2007.

Oakes, M., Ingall, E. D., Lai, B., Shafer, M. M., Hays, M. D., Liu, Z. G., Russell, A. G. and Weber, R. J.: Iron solubility related to particle sulfur content in source emission and ambient fine particles, Environ. Sci. Technol., 46(12), 6637–6644, doi:10.1021/es300701c, 2012.

Olgun, N., Duggen, S., Croot, P. L., Delmelle, P., Dietze, H., Schacht, U., Óskarsson, N., Siebe, C., Auer, A. and Garbe-Schönberg, D.: Surface ocean iron fertilization: The role of airborne volcanic ash from subduction zone and hot spot volcanoes and related iron fluxes into the Pacific Ocean, Global Biogeochem. Cycles, 25(GB4001), 1–15, doi:10.1029/2009GB003761, 2011.

Panias, D., Taxiarchou, M., Paspaliaris, I. and Kontopoulos, A.: Mechanisms of dissolution of iron oxides in aqueus oxalions, Hydrometallurgy, 42(95), 257–265, 1996.

Paris, R., Desboeufs, K. V., Formenti, P., Nava, S. and Chou, C.: Chemical characterisation of iron in dust and biomass burning aerosols during AMMA-SOP0/DABEX: Implication for iron solubility, Atmos. Chem. Phys., 10(9), 4273–4282, doi:10.5194/acp-10-4273-2010, 2010.

Paris, R., Desboeufs, K. V. and Journet, E.: Variability of dust iron solubility in atmospheric waters: Investigation of the role of oxalate organic complexation, Atmos. Environ., 45(36), 6510–6517, doi:10.1016/j.atmosenv.2011.08.068, 2011.

Perlwitz, J. P., Pérez García-Pando, C. and Miller, R. L.: Predicting the mineral composition of dust aerosols - Part 1: Representing key processes, Atmos. Chem. Phys., 15(20), 11593–11627, doi:10.5194/acp-15-11593-2015, 2015a.

Perlwitz, J. P., Pérez García-Pando, C. and Miller, R. L.: Predicting the mineral composition of dust aerosols - Part 2: Model evaluation and identification of key processes with observations, Atmos. Chem. Phys., 15(20), 11629–11652, doi:10.5194/acp-15-11629-2015, 2015b.

Price, C. A.: Iron compounds and plant nutrition, Annu. Rev. Plant Physiol., 19, 239–248, 1968.

Reddington, C. L., Spracklen, D. V., Artaxo, P., Ridley, D. A., Rizzo, L. V. and Arana, A.: Analysis of particulate emissions from tropical biomass burning using a global aerosol model and long-term surface observations, Atmos. Chem. Phys., 16(17), 11083–11106, doi:10.5194/acp-16-11083-2016, 2016.

Rémy, S., Veira, A., Paugam, R., Sofiev, M., Kaiser, J. W., Marenco, F., Burton, S. P.,
Benedetti, A., Engelen, R. J., Ferrare, R. and Hair, J. W.: Two global data sets of daily fire
emission injection heights since 2003, Atmos. Chem. Phys., 17(4), 2921–2942,
doi:10.5194/acp-17-2921-2017, 2017.
Ridley, D. A., Heald, C. L., Kok, J. F. and Zhao, C.: An observationally constrained estimate of
global dust aerosol optical depth, Atmos. Chem. Phys., 16(23), 15097–15117, doi:10.5194/acp-

1472   16-15097-2016, 2016.

Rogan, N., Achterberg, E. P., Le Moigne, F. A. C., Marsay, C. M., Tagliabue, A. and Williams,
R. G.: Volcanic ash as an oceanic iron source and sink, Geophys. Res. Lett., 43(6), 2732–2740,
doi:10.1002/2016GL067905, 2016.
Scanza, R. A., Mahowald, N., Ghan, S., Zender, C. S., Kok, J. F., Liu, X., Zhang, Y. and Albani,
S.: Modeling dust as component minerals in the Community Atmosphere Model: Development
of framework and impact on radiative forcing, Atmos. Chem. Phys., 15(1), 537–561,
doi:10.5194/acp-15-537-2015, 2015.
Scanza, R. A., Hamilton, D. S., Perez Garcia-Pando, C., Buck, C., Baker, A. and Mahowald, N.
M.: Atmospheric Processing of Iron in Mineral and Combustion Aerosols: Development of an
Intermediate-Complexity Mechanism Suitable for Earth System Models, Atmos. Chem. Phys.,
18, 14175–14196, doi:10.5194/acp-18-14175-80, 2018.
Schroth, A. W., Crusius, J., Sholkovitz, E. R. and Bostick, B. C.: Iron solubility driven by
speciation in dust sources to the ocean, Nat. Geosci., 2(5), 337–340, doi:10.1038/ngeo501,

1486   2009.

Schutgens, N., Tsyro, S., Gryspeerdt, E., Goto, D., Weigum, N., Schulz, M. and Stier, P.: On the
spatio-temporal representativeness of observations, Atmos. Chem. Phys., 17(16), 9761–9780,
doi:10.5194/acp-17-9761-2017, 2017.
Shi, Z., Krom, M. D., Jickells, T. D., Bonneville, S., Carslaw, K. S., Mihalopoulos, N., Baker, A.
R. and Benning, L. G.: Impacts on iron solubility in the mineral dust by processes in the source
region and the atmosphere: A review, Aeolian Res., 5(May), 21–42,
doi:10.1016/j.aeolia.2012.03.001, 2012.
Shoenfelt, E. M., Sun, J., Winckler, G., Kaplan, M. R., Borunda, A. L., Farrell, K. R., Moreno, P.
I., Gaiero, D. M., Recasens, C., Sambrotto, R. N. and Bostick, B. C.: High particulate iron(II)
content in glacially sourced dusts enhances productivity of a model diatom, Sci. Adv., 3(6), 1–
10, doi:10.1126/sciadv.1700314, 2017.
Sholkovitz, E. R., Sedwick, P. N., Church, T. M., Baker, A. R. and Powell, C. F.: Fractional
solubility of aerosol iron: Synthesis of a global-scale data set, Geochim. Cosmochim. Acta, 89,
173–189, doi:10.1016/j.gca.2012.04.022, 2012.
Smith, M. B., Mahowald, N. M., Albani, S., Perry, A., Losno, R., Qu, Z., Marticorena, B., Ridley,
D. A. and Heald, C. L.: Sensitivity of the interannual variability of mineral aerosol simulations to
meteorological forcing dataset, Atmos. Chem. Phys., 17(5), 3253–3278, doi:10.5194/acp-17-

1504    3253-2017, 2017.

Sofiev, M., Ermakova, T. and Vankevich, R.: Evaluation of the smoke-injection height from wild-
land fires using remote-sensing data, Atmos. Chem. Phys., 12(4), 1995–2006, doi:10.5194/acp-

1507    12-1995-2012, 2012.

Solmon, F., Chuang, P. Y., Meskhidze, N. and Chen, Y.: Acidic processing of mineral dust iron
by anthropogenic compounds over the north Pacific Ocean, J. Geophys. Res., 114(D2),
D02305, doi:10.1029/2008JD010417, 2009.
Strong, C. L., Bullard, J. E., Dubois, C., McTainsh, G. H. and Baddock, M. C.: Impact of wildfire
on interdune ecology and sediments: An example from the Simpson Desert, Australia, J. Arid
Environ., 74(11), 1577–1581, doi:10.1016/j.jaridenv.2010.05.032, 2010.
Teixeira, J. C., Carvalho, A. C., Tuccella, P., Curci, G. and Rocha, A.: WRF-chem sensitivity to
vertical resolution during a saharan dust event, Phys. Chem. Earth, 94, 188–195,
doi:10.1016/j.pce.2015.04.002, 2016.
Tobo, Y., Adachi, K., DeMott, P. J., Hill, T. C. J., Hamilton, D. S., Mahowald, N. M., Nagatsuka,
N., Ohata, S., Uetake, J., Kondo, Y. and Koike, M.: Glacially sourced dust as a potentially
significant source of ice nucleating particles, Nat. Geosci., 12(April), 253–258,
doi:10.1038/s41561-019-0314-x, 2019.
Turquety, S., Logan, J. A., Jacob, D. J., Hudman, R. C., Leung, F. Y., Heald, C. L., Yantosca, R.
M., Wu, S., Emmons, L. K., Edwards, D. P. and Sachse, G. W.: Inventory of boreal fire
emissions for North America in 2004: Importance of peat burning and pyroconvective injection,
J. Geophys. Res. Atmos., 112(12), 1–13, doi:10.1029/2006JD007281, 2007.
Ulery, A. L. and Graham, R. C.: Forest Fire Effects on Soil Color and Texture, Soil Sci. Soc. Am.
J., 57(1), 135–140, doi:10.2136/sssaj1993.03615995005700010026x, 1993.
Wagenbrenner, N. S., Chung, S. H. and Lamb, B. K.: A large source of dust missing in
Particulate Matter emission inventories? Wind erosion of post-fire landscapes, Elem Sci Anth,
5(2), 1–10, doi:10.1525/elementa.185, 2017.
Wagner, R., Jähn, M. and Schepanski, K.: Wildfires as a source of airborne mineral dust -
Revisiting a conceptual model using large-eddy simulation (LES), Atmos. Chem. Phys., 18(16),
11863–11884, doi:10.5194/acp-18-11863-2018, 2018.
Wang, R., Balkanski, Y., Boucher, O., Bopp, L., Chappell, A., Ciais, P., Hauglustaine, D.,
Peñuelas, J. and Tao, S.: Sources, transport and deposition of iron in the global atmosphere,
Atmos. Chem. Phys., 15(11), 6247–6270, doi:10.5194/acp-15-6247-2015, 2015.
Ward, D. E. and Hardy, C. C.: Smoke emission from wildland fires, Environ. Int., 17(2–3), 117–

1537 137, 1991.

Ward, D. S., Kloster, S., Mahowald, N. M., Rogers, B. M., Randerson, J. T. and Hess, P. G.:
The changing radiative forcing of fires: Global model estimates for past, present and future,
Atmos. Chem. Phys., 12(22), 10857–10886, doi:10.5194/acp-12-10857-2012, 2012.
Weber, R. J., Guo, H., Russell, A. G. and Nenes, A.: High aerosol acidity despite declining
atmospheric sulfate concentrations over the past 15 years, Nat. Geosci., 9(April), 1–5,
doi:10.1038/NGEO2665, 2016.
Wu, C., Liu, X., Diao, M., Zhang, K., Gettelman, A., Lu, Z., Penner, J. E. and Lin, Z.: Direct
comparisons of ice cloud macro- and microphysical properties simulated by the Community
Atmosphere Model version 5 with HIPPO aircraft observations, Atmos. Chem. Phys., 17(7),
4731–4749, doi:10.5194/acp-17-4731-2017, 2017.
Wu, M., Liu, X., Zhang, L., Wu, C., Lu, Z., Ma, P. L., Wang, H., Tilmes, S., Mahowald, N.,
Matsui, H. and Easter, R. C.: Impacts of Aerosol Dry Deposition on Black Carbon Spatial
Distributions and Radiative Effects in the Community Atmosphere Model CAM5, J. Adv. Model.
Earth Syst., 10(5), 1150–1171, doi:10.1029/2017MS001219, 2018.
Xu, N. and Gao, Y.: Characterization of hematite dissolution affected by oxalate coating, kinetics
and pH, Appl. Geochemistry, 23(4), 783–793, doi:10.1016/j.apgeochem.2007.12.026, 2008.
Yamasoe, M. A., Artaxo, P., Miguel, A. H. and Allen, A. G.: Chemical composition of aerosol
particles from direct emissions of vegetation fires in the Amazon Basin: Water-soluble species
and trace elements, Atmos. Environ., 34(10), 1641–1653, doi:10.1016/S1352-2310(99)00329-5,

1557  2000.

Zender, C. S., Bian, H. and Newman, D.: Mineral Dust Entrainment and Deposition (DEAD)
model: Description and 1990s dust climatology, J. Geophys. Res., 108(D14),
doi:10.1029/2002JD002775, 2003.
Zhang, Y., Mahowald, N., Scanza, R. A., Journet, E., Desboeufs, K., Albani, S., Kok, J. F.,
Zhuang, G., Chen, Y., Cohen, D. D., Paytan, A., Patey, M. D., Achterberg, E. P., Engelbrecht, J.
P. and Fomba, K. W.: Modeling the global emission, transport and deposition of trace elements
associated with mineral dust, Biogeosciences, 12(19), 5771–5792, doi:10.5194/bg-12-5771-
1565  2015, 2015.

Zhu, X., Prospero, J. and Millero, F.: Diel variability of soluble Fe(II) and soluble total Fe in North
Africa dust in the trade winds at Barbados, J. Geophys. Res., 102(7), 21297–21305, 1997.
Zhuang, G., Yi, Z., Duce, R. A. and Brown, P. R.: Link between iron and sulphur cycles
suggested by detection of Fe(n) in remote marine aerosols, Nature, 355(6360), 537–539,
doi:10.1038/355537a0, 1992.