# Peer review of "Improved methodologies for Earth system modelling of atmospheric soluble iron and"

_Geoscientific Model Development, 2019_

## Referee Comment (RC1) · Anonymous Referee #1 · 7 Jun 2019

Summary

This study presents a description of a new atmospheric iron dissolution scheme (MIMI) and compares the simulations with available cruise-based observations in the literature. A comprehensive statistical analysis of the model comparison with the observations is also presented. The authors further indicate the difference between the iron solubility calculated at each time step in the model versus the offline one which is routinely presented in most atmospheric Fe modeling studies and used in ocean models. I find this an important finding of the current work on the importance of using online parameterizations in Earth System Models that should be highlighted more in the text. The manuscript is very well-written and covers all aspects of current atmospheric Fe-modelling research. Some minor points, however, could be addressed before the final publication in GMD, in order to help the reader to better understand the model developments considered for this work.

Minor comments

As it is stated in the abstract, the MIMI is developed for use within Earth system models. Since Earth System modeling is characterized by a heavy computational burden in simulating atmospheric processes, it would be better to present some statistics and discuss more in the manuscript on how much computationally expensive the new module is (e.g., MIMI compared to the previous configuration or to the simple representation of soluble Fe - such as using offline Fe solubility distributions on dust deposited aerosols in the ocean (e.g., see Aumont et al., GMD, 2015, doi:10.5194/gmd-8-2465-2015), as well as on how many species are required to be implemented in the model, etc.

P.8 line 221: It is stated that Fe emissions come from all eight mineral dust species. However, at the beginning of the Sect. 2.3.1 it is provided the Fe-fraction for 5 dust species. Please explain.

Table 3. Is the med-soluble Fe the readily released Fe reported in Scanza et al. (2018)? If this fraction represents the initial solubility of Fe-containing dust species, why do the authors refer to it as "medium" soluble?

P.9 line 239: The authors state that fire iron emissions were globally scaled by a uniform factor of two. However, afterward, they stated that "Total iron emissions from fires in MIMI were 2.2 Tg a-1 Fe, representing an approximate increase in iron emissions from fires of around 25% compared with those from BAM-Fe (see P.10 line 269)". Is this because of the different BC fire inventories used in the models? Please explain.

Section 2.3.3: The authors do not state the total iron emissions from anthropogenic combustion sources as in the case of fire iron emissions (Sect. 2.3.2; line 232). Do the authors consider ship oil-combustion emission? If yes, do they apply the same initially solubility (i.e., 4%) and in what sizes? For completeness, it would be also useful to refer to the coarse fraction of anthropogenic combustion iron (if any).

Section 2.4: It is not clear in the manuscript how the model calculates SO4 and SOA; i.e., the proxies of H+ and oxalate concentrations for the iron dissolution scheme. Please give more details on their budget terms. Do the authors apply modal aerosol microphysics for SO4 and SOA size distributions, and if yes, how? Please discuss.

In the manuscript, it is stated that the pH is lowered to 1 (from 2) in the Aitken and the accumulation modes (line 319). Since, the aerosol pH range is a very important player in atmospheric iron processing, how the new parameterization compares with the previous one? Did you tune the model to match the previous configuration or to observations? How much is the iron dissolution production (per mode) and how does it compare to the previous model set-up in total soluble Fe production terms? Moreover, how does it compare now to other studies? Where is possible please provide figures.

Page 12; line 323: The authors state that "in-cloud organic dissolution reaction only occurs where cloud-borne aerosols are present". Do the authors mean the SO4 and oxalate production? What other cloud-borne aerosols the model considers?

Since oxalate is produced in the aqueous phase of the atmosphere, in contrast to other SOA that can be also produced via gas-to-particle partition, how the authors parameterized oxalate production in cloud droplets? Do you take into account the cloud fraction (and/or the presence of liquid water content?) in your calculations using SOA as a proxy? Please discuss. Moreover, how oxalate concentrations in the model are compared with other studies and with atmospheric observations. If possible, please provide the oxalate distributions of the model for the new parameterization as well as a comparison with observations.

Table 5: Please also provide the K(T), m and A values used for this study for each Fe-containing mineral.

Section 3.1: Please provide dust emission strengths per mode used in the model (also in Table 6 as for the other species). Since it is stated that dust lifetime has decreased, does this mean that the increase in dust emissions affect more the coarse mode of dust? Please discuss.

Page 20; line 487. Please provide statistics for the calculated improvement in the model.

Page 26; line 610: The authors state that emissions of dust are ∼80% higher and the iron it contains ∼120% higher in MIMI compared to those in BAM-Fe. What is the mean Fe/Dust fraction in the model after the applied corrections for the new model configuration? How is it compared to other studies?

Section 4.3; lines 662-663: the authors state that although in MIMI the amount of total iron deposited in the ocean is approximately double that estimated in BAM-Fe, the soluble iron deposition is similar (Table 7). As a reason for this, it is indicated the reduction in North Central Atlantic. Can this be also due to the different aerosol distribution considered in the models (i.e., bulk vs. modal) or the differences in the iron dissolution terms? Please discuss your conclusions and possibly provide a figure with relative differences between the MIMI and MAM-Fe.

Table 7: Although the percent contribution from combustion iron to total iron deposition after the correction in the model increases for all Northern and equatorial oceanic regions for MIMI compared to BAM-Fe, that is not the case for S. Atlantic, S. Pacific, S. Indian, and the Antarctic. Is this because of the increase of total iron due to the increase of dust deposited aerosols? Due to the different size distribution between the two versions of the model? Due to the different fire emission inventory? Or due to shipping emissions?

In page 5, line 151: "pre-cursor" to "precursor"

---

## Referee Comment (RC2) · Anonymous Referee #2 · 13 Jun 2019

In this paper, the authors address a very challenging aspect of global atmospheric modeling: emissions, fate, and transport of atmospheric iron. Authors present a comprehensive explanation of the new MIMI approach and compare model responses with results from the implementation of the preceding model (BAM-Fe). Among the updated MIMI inputs and processes were spatial distributions of fire and combustion emissions of iron; consideration of black carbon emissions to estimate iron contributions from fires; pH and emissions modifications with respect to aerosol mode; separate treat-

ment of interstitial and cloud processing; and speed of solubility. The results indicated that the combined updates improved model agreeability with observations in select regions. Comparison to BAM-Fe results suggest that previous modeled estimates of emissions and deposition of total iron were greatly underestimated.

In general, I appreciated the thorough consideration of many of the drivers of iron modeling uncertainty in this paper, as well as the transparent discussion of the issues that still need to be addressed to further improve iron representation in global models. Please see the following general comments and specific points that, if addressed, would potentially strengthen the paper.

1. Lack of observations and sensitivity to averaging are cited as sources of uncertainty in evaluating modeled soluble fraction. Is it possible that other drivers are important here? For example, does the presence or absence of other chemical species, or incorrect species distributions in the model, affect modeled iron solubility? In addition to evaluating sensitivity to averaging techniques, it makes sense to evaluate soluble fraction sensitivity to emissions of other soluble species.

2. The explanation of results throughout the paper would benefit from the inclusion of additional quantitative information. While the figures are very comprehensive, highlighting more quantitative outcomes within the text would strengthen the paper.

3. Table 3: What is the fate of the remaining fraction of each mineral treated in the model?

4. Table 4: Indicate in the header that these are fire emissions ratios.

5. Line 344: Should this be statistically?

6. Line 420, Figure 2: It would be more informative to include an additional table of slopes, intercepts, etc. for each region and for all regions combined.

7. Figure 4: Label the scatter plots as mean and standard deviation.

8. Line 547: "...differences between method are not insignificant..."

9. Lines 567-572: Repeated text.

10. Lines 572-574: As written, this sentence could be interpreted as, the ratios of tails only exist in certain regions. Whether narrow or wide, many distributions will have tails. Perhaps rewriting the sentence to indicate that extreme ratios of tails are found in specific regions would eliminate ambiguity.

11. Line 610: "...the iron it contains is ∼120% higher..."

12. Lines 616-617: This designation of fire emissions as combustion emissions here is inconsistent with the emissions categories presented in the rest of the paper.

13. Line 690: The first instance of acronym should be spelled out.

14. Lines 692-694: Was the sensitivity to vertical resolution near the surface tested in this study? If not, please cite a reference here.

15. Line 709: The first instance of acronym should be spelled out.

16. Section 5: This was by far the most well-written section of the paper. I found the writing of the majority of the other sections to be choppy and difficult to read.
* * *

---

## Author Comment (AC1) · 29 Jul 2019

We thank the reviewer for their thoughtful comments which have improved the manuscript. Line numbers refer to those in GMDD manuscript. New manuscript text is italicized in the replies.

**Comment:** This study presents a description of a new atmospheric iron dissolution scheme (MIMI) and compares the simulations with available cruise-based observations in the literature. A comprehensive statistical analysis of the model comparison with the observations is also presented. The authors further indicate the difference between the iron solubility calculated at each time step in the model versus the offline one which is routinely presented in most atmospheric Fe modeling studies and used in ocean models. I find this an important finding of the current work on the importance of using online parameterizations in Earth System Models that should be highlighted more in the text.

**Response:** We agree that the difference when using online results is an important finding and have thus brought it forward to now be included within the abstract at L32 as follows,

*"Comparison of iron solubility calculated at each model time step versus that calculated based on a ratio of the monthly mean values, which is routinely presented in aerosol studies and used in ocean biogeochemistry models, are on average globally one-third (34%) higher."*

And in the results at end of first paragraph of Section 3.4,

*"Overall, global annual mean iron solubility calculated online is one-third (34%; NH=40%, SH=29%) higher than when calculated offline."*

and strengthened this point within the conclusions at L895 as follows,

"There are significant differences in calculating iron solubility based on the order of the averaging operation. *When calculating at each model time-step global annual mean iron solubility is one-third (34%; NH=40%, SH=29%) higher than when calculated from monthly mean values. Earth-system models are designed to integrate land-atmosphere-ocean-ice components at each time-step and thus could yield different results based on the coupling time-step length employed. Furthermore, t*he mean […]"

**Comment:** MIMI is developed for use within Earth system models. Since Earth System modeling is characterized by a heavy computational burden in simulating atmospheric processes, it would be better to present some statistics and discuss more in the manuscript on how much computationally expensive the new module is (e.g., MIMI compared to the previous configuration or to the simple representation of soluble Fe - such as using offline Fe solubility distributions on dust deposited aerosols in the ocean (e.g., see Aumont et al., GMD, 2015, doi:10.5194/gmd-8-2465-2015), as well as on how many species are required to be implemented in the model, etc.

**Response:** We agree that this is informative for future development in other models. New methods text added in new Section 2.4.2,

"*2.4.2 Computational costs*

*Earth System models are generally characterized by having a heavy computational burden in simulating atmospheric processes. The inclusion of MIMI requires eight dust mineral tracers (a net addition of seven) and six iron tracers. The total addition of aerosol tracers new is 39 (13 in each of the three aerosol modes) if dust minerology is not already present, or 18 new aerosol tracers if it is (e.g., NASA GISS model (Perlwitz et al., 2015b, 2015a)). The additional computational cost of MIMI within CESM-CAM5 is approximately a doubling of the required core-hours; around half of that is associated with dust minerology speciation and the other half with iron speciation and processing (Table 6) Note that additional computational tuning, or changes in configuration, could modify these computational change estimates. For example, with dust minerology (MAM4DU8) there is an approximate 3-fold increase in required core-hours due to model structural differences when transitioning from CAM5 to CAM6.*"

Additional discussion relating to Reviewers comment about using offline estimates of iron solubility added at L476 within Section 3.3,

"*In the absence of iron atmospheric process modelling, ocean biogeochemistry models with an iron component  (e.g., Aumont et al., 2015; Moore et al., 2004) have estimated iron solubility from offline dust modelling by means of an assumption that it contains 3.5% iron by weight, of which 2% is soluble. Iron solubility is highly temporally and spatially*

*variable however, and in the absence of spatial atmospheric emission information, pyrogenic iron sources, and atmospheric processing of iron an estimate of 2% solubility leads to underestimates of observed iron solubility in nearly all HNLC ocean regions (Figure 4).*"

New Table 6:

*"Table 6: Simulation time (in seconds per simulated year) for the CESM-MAM4 model. The CAM5 base model, with the addition of dust minerology, and with the addition of dust minerology and iron processing (i.e., MIMI v1.0) shown in black text. Cost of running the new higher resolution CAM6 model with dust minerology also shown for comparison in blue text. All CAM5 simulations executed on 10 nodes, with 36 cores per node, for two years (2006-2007) with consistent output fields.*"

|  | CAM5 | | | CAM6 |
|---|---|---|---|---|
|  | MAM4 (Base model) | MAM4DU8 (dust minerology) | MAM4DU8FE6 (MIMIv1.0) | MAM4DU8 (dust minerology) |
| Number advected aerosol species | 24 | 45 | 63 | 46 |
| Gridcell resolution (#lon x #lat) | 144x96 | 144x96 | 144x96 | 288x192 |
| Wall clock s a$^{-1}$(simulation) | 3954 | 5856 | 7836 | 20167 |
| Core-hours | 396 | 586 | 784 | 2017 |

**Comment:** P.8 line 221: It is stated that Fe emissions come from all eight mineral dust species. However, at the beginning of the Sect. 2.3.1 it is provided the Fe-fraction for 5 dust species. Please explain.

**Response:** While MIMI carries 8 mineral dust species, only 5 of these provide iron. We have updated the manuscript to be clearer as follows:

"Iron emissions from *the five iron-bearing* mineral dust species *(three dust minerals contain no iron)* were then partitioned […]"

**Comment:** Table 3. Is the med-soluble Fe the readily released Fe reported in Scanza et al. (2018)? If this fraction represents the initial solubility of Fe-containing dust species, why do the authors refer to it as "medium" soluble?

**Response:** The medium soluble iron tracer is the sum of the readily released ("fast") iron (soluble fraction at emission) and the iron which is created via medium reacting atmospheric processing (additional soluble fraction created during transport). As Table 3 is referring to emissions we now add additional text to the caption to be clearer,

"*At emission med-soluble iron is equivalent to the fast-soluble iron fraction (i.e., the fraction which is already assumed to be soluble at emission).*"

And in the main text at L224 as follows,

"Note that, slow *and med*-soluble iron are only produced by *non-reversable* atmospheric processing within the model*; therefore, computational costs can be reduced by not creating a separate iron tracer representing the fraction which is already soluble at emission (i.e., 'fast' reacting), but instead add an initial med-soluble iron processed emission burden which is equivalent to the assumed fast reacting iron fraction*."

**Comment:** The Authors state that fire iron emissions were globally scaled by a uniform factor of two. However, afterward, they stated that "Total iron emissions from fires in MIMI were 2.2 Tg a-1 Fe, representing an approximate increase in iron emissions from fires of around 25% compared with those from BAM-Fe (see P.10 line 269)". Is this because of the different BC fire inventories used in the models? Please explain.

**Response:** The reviewer is correct that this is due to different fire inventories between models. We have further highlighted this for the reader on L270 as follows,

"*The lower 25% increase between BAM-Fe and MIMI iron emissions, as compared to the doubling of the fire iron emissions themselves within MIMI, is due to different underlying fire emission inventories used in each model.*"

**Comment:** Section 2.3.3: The authors do not state the total iron emissions from anthropogenic combustion sources as in the case of fire iron emissions (Sect. 2.3.2; line

232). Do the authors consider ship oil-combustion emission? If yes, do they apply the same initially solubility (i.e., 4%) and in what sizes?

**Response:** We do not consider shipping emissions in this version of the model.

**Comment:** For completeness, it would be also useful to refer to the coarse fraction of anthropogenic combustion iron (if any).

**Response:** We have added the following text at L284 as suggested,

*"Resulting fine mode anthropogenic combustion emissions were 0.50 Tg Fe a$^{-1}$ and coarse mode emissions were 2.8 Tg Fe a$^{-1}$. Similar to fire emissions, 10% of fine size emissions were partitioned into the Aiken mode at emission, the remainder 90% of fine size emissions were emitted into the accumulation mode, and 100% of coarse size emissions were emitted to the coarse mode."*

**Comment:** Section 2.4: It is not clear in the manuscript how the model calculates SO4 and SOA; i.e., the proxies of H+ and oxalate concentrations for the iron dissolution scheme. Please give more details on their budget terms.

**Response:** Sulfate and SOA aerosol are fundamental components of the host CAM model (and all aerosol models) and thus have been described in detail elsewhere in the literature. We therefore feel it best to refer readers to this literature for detailed information but include the basics in the text and plots of simulated burden of each to the SI. New text at L301 in Section 2.4,

*"The proton dissolution scheme was dependent upon an estimated [H+], calculated from the ratio of sulphate to calcite, and the simulated temperature. […] Both the sulphate and secondary organic carbon aerosol (Fig. S1), upon which the iron processing requires, are fundamental components of aerosol models (e.g., Kanakidou et al., 2005; Mann et al., 2014). In CAM sulfate is mainly formed via oxidation of SO$_{2(aq)}$ with a smaller contribution from H$_2$SO$_4$ condensation on aerosol while secondary organic aerosol is formed via the partitioning of semi-volatile organic gases (Liu et al., 2012). Neither gas-to-particle production processes are structurally modified from the description of CAM5 by Lui et al. (2012, 2016) by the incorporation of MIMI. A structural model improvement […]"*

[Figure]

**Figure S1.** Atmospheric annual mean column burden of sulfate and secondary organic aerosol for year 2007.

**Comment:** Do the authors apply modal aerosol microphysics for SO4 and SOA size distributions, and if yes, how? Please discuss.

**Response:** Yes, all aerosol species undergo aerosol microphysical processes and are size resolved. But, as stated above, these aerosol species are integral components of the host aerosol model and its description is beyond this paper. We therefore briefly increase our outline of the host model (in Section 2) and refer interested readers to the detailed model description papers as follows on L147,

"The other major aerosol species […] *However, atmospheric iron processing in MIMI requires both sulphate and (secondary) organic aerosols to be simulated as they act as proxies for the reactant species of [H⁺] and oxalate, respectively. In CAM5 sulphate aerosol is present in all three aerosol modes while secondary organic aerosol is only present in the fine Aitken and accumulation modes (Liu et al., 2012, 2016). Aerosol microphysics was applied in the same way to the new iron aerosol tracers as the base aerosol species (Liu et al., 2012, 2016).*"

**Comment:** Since, the aerosol pH range is a very important player in atmospheric iron processing, how the new parameterization compares with the previous one? Did you tune

the model to match the previous configuration or to observations? How much is the iron dissolution production (per mode) and how does it compare to the previous model set-up in total soluble Fe production terms? Moreover, how does it compare now to other studies? Where is possible please provide figures.

**Response:** We cover aspects of this and the related next two oxalate comments within Section 4.2: Iron atmospheric processing comparison. We feel it best to continue this discussion there (rather than in the methods) and point the reader at the end of the pH paragraph in section 2.4 as follows,

"*See Section 4.2 for comparison of acid processing in MIMI with literature and previous model (BAM-Fe).*"

Figure 10 compares the distribution of the previous model pH with the new version and is described in L631-637. During development aerosol pH was also compared to published ISORROPIA II results. We added new text incorporating this missing detail as follows:

"*Comparison of Fig 10. to modelled pH estimates by Myriokefalitakis et al. (2015) shows generally good agreement in the NH, but in the SH MIMI simulates less acidic coarse mode aerosol over continental regions and more acidic aerosol over marine regions. As iron models are unable to capture the high observed iron solubility (>10%) over SH marine regions (Myriokefalitakis et al., 2018), and in the absence of remote pH aerosol observations, we suggest that our basic parameterization captures an aerosol pH which is suitable for use in Earth system models.*"

In order to directly compare with the literature (included previous BAM-Fe results) we follow (Myriokefalitakis et al., 2018) and sum iron dissolution with soluble iron emissions (i.e., the net soluble iron source) to create two new maps as follows,

[Figure]

**Figure 12.** Annual mean dust and pyrogenic (sum of fires and anthropogenic combustion) soluble iron source (i.e., sum of emissions and atmospheric processing).

New text,

*"Comparison of mineral dust and pyrogenic sources of modelled soluble iron (sum of emissions and atmospheric dissolution; Fig. 12) with the four iron models (including BAM-Fe) reported by Myriokefalitakis et al. (2018) shows that the spatial distribution in MIMI is broadly similar for most regions of the world. A notable difference exists in the North Pacific region where the soluble iron source in MIMI is lower than all other iron models, and similarly with total iron concentrations when compared to observations (Figs. 4 and 5). Future development of MIMI should thus be focused on the North Pacific, including the addition of shipping soluble iron emissions which are relatively concentrated in this region (Ito, 2013). An improvement for MIMI can be seen over the Atlantic region directly downwind of Saharan soluble iron sources. In general, iron models are over representing iron solubility close to dust sources compared to observations (Myriokefalitakis et al., 2018) and in order for BAM-Fe to reach better agreement with observed iron solubility in this region dust emissions of soluble iron had to be scaled downwards (Conway et al., 2019). We suggest this improvement is linked to the improved modal representation of aerosol pH in MIMI (Fig.10)."*

**Comment:** Page 12; line 323: The authors state that "in-cloud organic dissolution reaction only occurs where cloud-borne aerosols are present". Do the authors mean the $SO_4$ and

oxalate production? What other cloud-borne aerosols the model considers? Since oxalate is produced in the aqueous phase of the atmosphere, in contrast to other SOA that can be also produced via gas-to-particle partition, how the authors parameterized oxalate production in cloud droplets? Do you take into account the cloud fraction (and/or the presence of liquid water content?) in your calculations using as a proxy? Please discuss.

**Response:** In order to explain these issues better, we propose replacing said sentence with new text as follows,

"*All aerosol species in the host CAM5 framework are carried in either an interstitial (i.e., not associated with water) or cloud-borne (i.e., associated with water) phase. The organic-ligand reaction only proceeds within MIMI if the condition that cloud is present in the grid-cell is first met. If cloud is present then only the iron aerosol which is associated with water undergoes organic ligand processing (i.e., the interstitial aerosol component remains unchanged). Any future development of MIMI within an aerosol model which does not advect a separate tracer for the cloud-borne phase of aerosol would therefore need to adjust the reaction to take account of this.*"

**Comment:** Moreover, how oxalate concentrations in the model are compared with other studies and with atmospheric observations. If possible, please provide the oxalate distributions of the model for the new parameterization as well as a comparison with observations.

**Response:** Again, we first point the reader to Section 4.2 where the discussion element of the changes to atmospheric dissolution of iron occurs,

"S*ee Section 4.2 for comparison of in-cloud oxalate dissolution in MIMI with literature and previous model (BAM-Fe).*"

While Figure 11 already compares oxalate between MIMI and its previous version we have added new panel to Figure 11 of the oxalate comparison to observation from Myriokefalitakis et al. (2011) (Table S3).

[Figure]

**Figure 11: A:** Relative difference in organic ligand reaction on in-cloud iron aerosol dissolution between MIMI and BAM-Fe. Due to significant differences in simulated cloud cover between CAM4 and CAM5 oxalate concentrations [OXL] are multiplied by the model simulated cloud fraction in this figure. **B***: Surface level oxalate (OXL) concentration in the model and observations. Model values are annual mean (2007-2011) and monthly standard deviation. Observation values are from Table S3 in Myriokefalitakis et al. (2011) and reported with uncertainty where given.*

And the following new text

*"Compared to observations (Myriokefalitakis et al., 2011) modelled oxalate concentrations are well represented at high observed concentrations but are biased low when observed concentrations are low (Fig. 11b). The low model bias is stronger within remote observational regions (marine vs. urban observation sites), suggesting that the removal of secondary organic aerosol may be too strong within the model and/or that there is a missing marine aerosol pre-cursor gas emissions source (Facchini et al., 2008; O'Dowd and de Leeuw, 2007) in this model which significantly lowers simulated secondary organic aerosol, and thus oxalate, concentrations."*

**Comment:** Table 5: Please also provide the K(T), m and A values used for this study for each Fe-containing mineral.

**Response:** Added following values to table:

$$K_{med}(T) = 1.3 \times 10^{-11} \times e^{6.7 \times 10^3 \times \left(\frac{1.0}{298.0} - \frac{1.0}{temp(K)}\right)} \text{ moles m}^2 \text{ s}^{-1}$$

$$K_{slow}(T) = 1.8 \times 10^{-11} \times e^{9.2 \times 10^3 \times \left(\frac{1.0}{298.0} - \frac{1.0}{temp(K)}\right)} \text{ moles m}^2 \text{ s}^{-1}$$

$m_{med} = 0.39$

$m_{slow} = 0.50$

$A_{med} = 90.0 \text{ m}^2 \text{ g}^{-1}$

$A_{slow} = 100.0 \text{ m}^2 \text{ g}^{-1}$

**Comment:** Section 3.1: Please provide dust emission strengths per mode used in the model (also in Table 6 as for the other species). Since it is stated that dust lifetime has decreased, does this mean that the increase in dust emissions affect more the coarse mode of dust? Please discuss.

**Response:** Added following modal emissions to text on L421 and Table 6:

Aiken = 16 Tg a$^{-1}$, accumulation = 36 Tg a$^{-1}$, coarse = 3198 Tg a$^{-1}$.

New text on L424:

"[…] because dust lifetime has proportionally decreased (Table S2) *which affects coarse mode dust aerosol (where 98 – 99% of total dust mass is emitted) more than fine mode dust aerosol*."

**Comment:** Page 26; line 610: The authors state that emissions of dust are ~80% higher and the iron it contains~120% higher in MIMI compared to those in BAM-Fe. What is the mean Fe/Dust fraction in the model after the applied corrections for the new model configuration? How is it compared to other studies?

**Response:** We thank the Reviewer for this insightful suggestion. New text describing the Fe/Dust fraction,

*"The simulated annual mean iron in dust percentage is 4.1%, with the highest percent occurring in the coarse mode at 6.5% and lowest percent occurring in the Aiken mode at 1.1%. Accounting for dust minerology therefore increases the global mean iron percent by weight above the currently well-used global mean estimate of 3.5% (e.g., Jickells et al., 2005; Shi et al., 2012)."*

**Comment:** Section 4.3; lines 662-663: the authors state that although in MIMI the amount of total iron deposited in the ocean is approximately double that estimated in BAM-Fe, the soluble iron deposition is similar (Table 7). As a reason for this, it is indicated the reduction in North Central Atlantic. Can this be also due to the different aerosol distribution considered in the models (i.e., bulk vs. modal) or the differences in the iron dissolution terms? Please discuss your conclusions and possibly provide a figure with relative differences between the MIMI and MAM-Fe.

**Response:** The Reviewer is correct to point out the structural differences between bulk and modal aerosol model can affect deposition (as shown by changes to dust lifetimes in Table S2) as well as differences in dissolution. The atmospheric dissolution comparisons are shown in Figs. 10 and 11 and the new Figure 12 suggested above further shows the source of soluble iron is significantly lower than CAM4 for this region. We therefore feel another figure is not necessary here. New text at L668,

*"In MAM4 dust is treated as internally mixed aerosol with sea salt, leading to higher rates of wet deposition than when dust is externally mixed aerosol (Liu et al., 2012) as it is in CAM4. The internally mixed treatment of dust aerosol in MAM4 is thus an important factor leading to the lower simulated dust lifetime when compared to BAM-Fe (Table S2). Over the North Central Atlantic region, the combination of a lower soluble iron source (Fig. 12 compared to Fig. S4b by Myriokefalitakis et al. (2018)), dust atmospheric lifetime (Table S2), lower aerosol pH (Fig. 10), and lower relative organic ligand processing (Fig 11) will all work towards reducing the magnitude of atmospheric soluble iron deposition flux in MAM4 compared to BAM-Fe."*

**Comment:** Table 7: Although the percent contribution from combustion iron to total iron deposition after the correction in the model increases for all Northern and equatorial oceanic regions for MIMI compared to BAM-Fe, that is not the case for S. Atlantic, S.

Pacific, S. Indian, and the Antarctic. Is this because of the increase of total iron due to the increase of dust deposited aerosols? Due to the different size distribution between the two versions of the model? Due to the different fire emission inventory? Or due to shipping emissions?

**Response:** As the Reviewer suggests there are multiple reasons which could result in this change. Without a series of dedicated sensitivity simulations, which are beyond this study, it cannot be quantified which is dominating. We therefore include the important differences that the Reviewer raises in terms of this change as follows:

**"***The percent contribution from pyrogenic iron to total iron deposition between MIMI and BAM-Fe is more similar for all northern and equatorial oceanic regions than southern oceanic regions. Beyond the correction to anthropogenic combustion emissions, which are NH dominated, could be due to differences in the emissions of both dust and fire aerosol, structural differences between models relating to the aerosol size and composition which alters aerosol deposition rates, or a lower soluble iron source (Fig. 12); it is most likely to be a combination of all three.***"*

**Comment:** In page 5, line 151: "pre-cursor" to "precursor"

**Response:** Changed here and also a second occurrence on L650.

[revised manuscript text omitted]

---

## Author Comment (AC2) · 29 Jul 2019

We thank the reviewer for their thoughtful comments which have improved the manuscript. New manuscript text is italicized in the replies.

**Comment:** 1. Lack of observations and sensitivity to averaging are cited as sources of uncertainty in evaluating modeled soluble fraction. Is it possible that other drivers are important here? For example, does the presence or absence of other chemical species, or in-correct species distributions in the model, affect modeled iron solubility? In addition to evaluating sensitivity to averaging techniques, it makes sense to evaluate soluble fraction sensitivity to emissions of other soluble species.

**Response:** There are other factors beyond the analytical method under which iron solubility is calculated causing uncertainty. We feel we have covered many of the most important points in a detailed manner within L493-500 and the dedicated Future directions Section 5. The undertaking of many more detailed sensitivity simulations is beyond the scope of this model description paper; but this is an excellent suggestion for a follow-on paper within a more relevant journal (e.g., ACP). We wish to convey the Authors comments though and add the following text in opening paragraph to Section 5,

*"In general, improving the modelled representation of secondary organic aerosol (including oxalate) and aerosol pH, particularly for remote regions, is an important task for aerosol modeling and one which have co-benefits for iron aerosol modelling. Comparisons of soluble fraction of other aerosol species with observations could also be used to guide model development."*

**Comment:** 2. The explanation of results throughout the paper would benefit from the inclusion of additional quantitative information. While the figures are very comprehensive, highlighting more quantitative outcomes within the text would strengthen the paper.

**Response:** We have made the manuscript more qualitative by including more information from Tables and Figures within the text.

**Comment:** 3. Table 3: What is the fate of the remaining fraction of each mineral treated in the model?

**Response:** The remaining mineral fractions are advected as their respective mineral species. New Table 6, as suggested by R1, helps further highlight this for the reader by including the number of advected tracers,

*"Table 6: Simulation time (in seconds per simulated year) for the CESM-MAM4 model. The CAM5 base model, with the addition of dust minerology, and with the addition of dust minerology and iron processing (i.e., MIMI v1.0) shown in black text. Cost of running the new higher resolution CAM6 model with dust minerology also shown for comparison in blue text. All CAM5 simulations executed on 10 nodes, with 36 cores per node, for two years (2006-2007) with consistent output fields."*

| | CAM5 | | | CAM6 |
| --- | --- | --- | --- | --- |
| | MAM4 (Base model) | MAM4DU8 (dust minerology) | MAM4DU8FE6 (MIMIv1.0) | MAM4DU8 (dust minerology) |
| Number advected aerosol species | 24 | 45 | 63 | 46 |
| Gridcell resolution (#lon x #lat) | 144x96 | 144x96 | 144x96 | 288x192 |
| Wall clock s a$^{-1}$(simulation) | 3954 | 5856 | 7836 | 20167 |
| Core-hours | 396 | 586 | 784 | 2017 |

and we also add the following text to the header of Table 3,

*"Residual mineral dust mass is then advected as its respective tracer."*

Note that this is covered in the main text already in L216-220 and so do not add any new text to the body of the manuscript.

**Comment:** 4. Table 4: Indicate in the header that these are fire emissions ratios.

**Response:** Added at end of header.

"Modelled fire emission ratio for Fe:BC then calculated from observed ratios."

**Comment:** 5. Line 344: Should this be statistically?

**Response:** Yes, thank you.

**Comment:** 6. Line 420, Figure 2: It would be more informative to include an additional table of slopes, intercepts, etc. for each region and for all regions combined.

**Response:** As Figure 2 has limited points per region (and so regional statistics would not be robust) we have included Hemispheric level details for Figure 2. However, we think this an excellent suggestion for a regional scale evaluation of total iron for Figure 4.

New text for Figure 2,

*Globally, both dust concentrations (correlation: r2 = 0.89) and deposition (correlation: $r^2$ = 0.83) are simulated well compared to observation within MIMI. A higher correlation of modelled dust concentrations with observations is calculated in the Northern Hemisphere (NH; $r^2$ = 0.89) compared to the Southern Hemisphere (SH; $r^2$ = 0.67), but with gradient of line of best fit is further from 1:1 (NH: 1.22 vs. SH: 1.07). Conversely, for dust deposition a lower correlation with observations is simulated in NH ($r^2$ = 0.75) compared to the SH ($r^2$ = 0.60) but with a gradient of the line of best fit closer to 1:1 (NH: 1.07 vs. SH: 0.72).*

Updated Figure 2:

[Figure]

"**Figure 2.** Dust aerosol optical depth, surface concentrations and deposition in modal aerosol model and observations (Albani et al., 2014; Holben et al., 2000). *Correlation ($r^2$), gradient (m) and intercept (c) shown for global (G), Northern Hemisphere (N) and Southern Hemisphere (S) regions.*"

Updated Figure 4:

[Figure]

**Figure 4.** Daily mean model total iron concentration and solubility from 2007 to 2011. Observations (circles) overlaid (at resolution of the model grid) as a mean from 1524 individual records in Mahowald et al. (2009) and in Myriokefalitakis et al. (2018). Also shown are scatter plots of the model mean and standard deviation compared to each available observation and identified by oceanic region. *Correlation ($r^2$), gradient (m) and intercept (c) for total iron with observations shown for each region.*

**Comment:** 7. Figure 4: Label the scatter plots as mean and standard deviation.

**Response:** Added (see above).

**Comment:** 8. Line 547: "...differences between method are not insignificant..."

**Response:** Altered as suggested.

**Comment:** 9. Lines 567-572: Repeated text.

**Response:** Removed repeated text.

**Comment:** 10. Lines 572-574: As written, this sentence could be interpreted as, the ratios of tails only exist in certain regions. Whether narrow or wide, many distributions will have tails. Perhaps rewriting the sentence to indicate that extreme ratios of tails are found in specific regions would eliminate ambiguity.

**Response:** Altered sentence as suggested, now reads as,

"The *extreme* ratio of the tails of soluble and total iron are only found *in specific* regions with the highest temporal variability […] "

**Comment:** 11. Line 610: "...the iron it contains is~120% higher..."

**Response:** Altered as suggested.

**Comment:** 12. Lines 616-617: This designation of fire emissions as combustion emissions here is inconsistent with the emissions categories presented in the rest of the paper.

**Response:** We agree that keeping consistency throughout is important to keep the reader orientated. We have removed the reference to combustion and now refer to the sum of fires and anthropogenic combustion as pyrogenic iron both here and throughout the manuscript where confusion could occur.

**Comment:** 13. Line 690: The first instance of acronym should be spelled out.

**Response:** Line 40 contains first instance of acronym and is spelled out.

**Comment:** 14. Lines 692-694: Was the sensitivity to vertical resolution near the surface tested in this study? If not, please cite a reference here.

**Response:** It was not and to our knowledge no study in the literature has explicitly examined this for dust deposition (but have for PM10 and the vertical profile). Removed previous sentenced and replaced with new text,

"*The dry deposition flux is sensitive to the aerosol properties, surface roughness and modelled turbulence. Although increasing the vertical resolution has been shown to increase surface $PM_{10}$ concentration (Menut et al., 2013) and better simualte the dust vertical profile (Teixeira et al., 2016), it is not as yet clear if this would correspondingly increase the dry deposition flux.*"

**Comment:** 15. Line 709: The first instance of acronym should be spelled out.

**Response:** This is the only instance of said acronym in the main text and so remove it and just state as follows,

"Inter Tropical Convergence Zone"

**Comment:** 16. Section 5: This was by far the most well-written section of the paper. I found the writing of the majority of the other sections to be choppy and difficult to read.

**Reply:** We have included many improvements to the text from both Reviewers and made additional ones ourselves where we could see it would help improve the manuscript.

References:

Menut, L., Bessagnet, B., Colette, A. and Khvorostiyanov, D.: On the impact of the vertical resolution on chemistry-transport modelling, Atmos. Environ., 67, 370–384,

doi:10.1016/j.atmosenv.2012.11.026, 2013.

Teixeira, J. C., Carvalho, A. C., Tuccella, P., Curci, G. and Rocha, A.: WRF-chem sensitivity to vertical resolution during a saharan dust event, Phys. Chem. Earth, 94, 188–195, doi:10.1016/j.pce.2015.04.002, 2016.